EMBO
Molecular Medicine

# Highly secreted tryptophanyl tRNA synthetase 1 as a potential theranostic target for hypercytokinemic severe sepsis

Yoon Tae Kim [1,16], Jin Won Huh [2,16], Yun Hui Choi [3], Hee Kyeong Yoon[3], Tram TT Nguyen[3], Eunho Chun[1], Geunyeol Jeong[1], Sunyoung Park[4], Sungwoo Ahn [5], Won-Kyu Lee [6] Young-Woock Noh[6], Kyoung Sun Lee[7], Hee-Sung Ahn [8], Cheolju Lee [9], Sang Min Lee[10] Kyung Su Kim[11], Gil Joon Suh[12], Kyeongman Jeon [13], Sunghoon Kim [14] & Mirim Jin [1,3,4,15] ✉

## Abstract

Despite intensive clinical and scientific efforts, the mortality rate of sepsis remains high due to the lack of precise biomarkers for patient stratification and therapeutic guidance. Secreted human tryptophanyl-tRNA synthetase 1 (WARS1), an endogenous ligand for Toll-like receptor (TLR) 2 and TLR4 against infection, activates the genes that signify the hyperinflammatory sepsis phenotype. High plasma WARS1 levels stratified the early death of critically ill patients with sepsis, along with elevated levels of cytokines, chemokines, and lactate, as well as increased numbers of absolute neutrophils and monocytes, and higher Sequential Organ Failure Assessment (SOFA) scores. These symptoms were recapitulated in severely ill septic mice with hypercytokinemia. Further, injection of WARS1 into mildly septic mice worsened morbidity and mortality. We created an anti-human WARS1-neutralizing antibody that suppresses proinflammatory cytokine expression in marmosets with endotoxemia. Administration of this antibody into severe septic mice attenuated cytokine storm, organ failure, and early mortality. With antibiotics, the antibody almost completely prevented fatalities. These data imply that blood-circulating WARS1-guided anti-WARS1 therapy may provide a novel theranostic strategy for life-threatening systemic hyperinflammatory sepsis.

Keywords Sepsis; Theranostics; Tryptophanyl-tRNA Synthetase (WARS1); Hypercytokinemia; Anti-WARS1 Antibody
Subject Categories Biomarkers; Immunology

## Introduction

Sepsis is a life-threatening clinical illness characterized by organ dysfunction resulting from a dysregulated immunological response to an infection (Singer et al, 2016). Alterations in circulation and metabolism can lead to septic shock (Hotchkiss et al, 2016; Singer et al, 2016), which can result in organ failure and death. Globally, the death rate for septic shock remains over 40% without specialized medicines (Rudd et al, 2020). Decades of research seeking biological agents that modify the immune responses to sepsis have been unable to demonstrate a significant positive effect or reduction in mortality (Cavaillon et al, 2020; Marshall, 2014). A possible explanation for this failure is that single cytokine antagonists are unable to suppress excessive systemic inflammation, which is biologically complicated, redundant, and difficult to ascertain in the early stages (Abraham et al, 1998; Abraham et al, 1995; Marshall, 2008). The other factor contributing to the lack of efficacy is largely the syndrome's heterogeneity (Leligdowicz and Matthay, 2019). A large-scale transcriptomic analysis of peripheral blood leukocytes across multiple sepsis cohorts, along with phenotype analysis using biochemical parameters, has revealed the sub-populations that can be classified into several endotypes. They exhibit distinct host immune response patterns; broadly, these are hyperinflammatory phenotypes that are primarily controlled by *RELB*, *NFKB2*, *REL*, and *IRF* vs. immune suppression phenotypes (Scicluna et al, 2017; Seymour et al, 2017; Sweeney et al, 2018). Hence, there has been a great deal of interest in precision immunotherapy targeting rationally categorized subgroups of patients with sepsis (Peters Van Ton et al, 2018). A recall analysis of a sepsis clinical trial evaluating the efficacy of rhIL-1RA revealed

[1]Department of Health Sciences and Technology, GAIHST, Gachon University, Incheon, Republic of Korea. [2]Department of Pulmonary and Critical Care Medicine, Asan Medical Center, University of Ulsan College of Medicine, Seoul, Republic of Korea. [3]R&D Center, MirimGENE, Incheon, Republic of Korea. [4]Lee Gil Ya Cancer and Diabetes Institute, Gachon University, Incheon, Republic of Korea. [5]Arnie Charbonneau Cancer Institute, University of Calgary, Calgary, AB, Canada. [6]New Drug Development Center, Osong Medical Innovation Foundation, Cheongju, Republic of Korea. [7]Non-Clinical Evaluation Center, Osong Medical Innovation Foundation, Cheongju, Republic of Korea. [8]Convergence Medicine Research Center, Asan Institute for Life Sciences, Asan Medical Center, Seoul, Republic of Korea. [9]Chemical & Biological Integrative Research Center, Korea Institute of Science and Technology, Seoul, Republic of Korea. [10]Department of Internal Medicine, Gil Medical Center, College of Medicine, Gachon University, Incheon, Republic of Korea. [11]Department of Emergency Medicine, Seoul National University Hospital, Seoul, Republic of Korea. [12]Department of Emergency Medicine, Seoul National University College of Medicine, Seoul, Republic of Korea. [13]Division of Pulmonary and Critical Care Medicine, Department of Medicine, Samsung Medical Center, Sungkyunkwan University School of Medicine, Seoul, Republic of Korea. [14]Medicinal Bioconvergence Research Center, Institute for Artificial Intelligence and Biomedical Research, The interdisciplinary graduate program in integrative biotechnology, College of Pharmacy & College of Medicine, Gangnam Severance Hospital, Yonsei University, Incheon, Republic of Korea. [15]Department of Microbiology, College of Medicine, Gachon University, Incheon, Republic of Korea. [16]These authors contributed equally: Yoon Tae Kim, Jin Won Huh. ✉E-mail: mirimj@gachon.ac.kr

that baseline plasma IL-1RA concentration had a significant influence on patient response; subjects stratified by high IL-1RA levels (likely under hyperinflammatory status) experienced a significant effect (Meyer et al, 2018), indicating that initial plasma IL-1RA levels may be used to stratify patients who can benefit from the anti-inflammatory therapy. In addition, a recent clinical investigation reported that the administration of anakinra to stratified sepsis patients with excessive inflammation increased 7-day survival (Leventogiannis et al, 2022). Hence, for sepsis precision immunotherapy, a theranostic biomarker that can identify patients based on their underlying immunopathologic condition while also functioning as a therapeutic target based on the biological mechanisms is required (Stanski and Wong, 2020).

Tryptophanyl tRNA synthetase 1 (WARS1), one of aminoacyl tRNA synthetases (ARSs), is classically known as an essential housekeeping enzyme that ligates tryptophan to its cognate tRNA for protein synthesis (Jin, 2019; Kwon et al, 2019). Interestingly, when there is a pathogenic infection, WARS1 is promptly released from monocytes into the extracellular space (Ahn et al, 2017; Nguyen et al, 2023). The secreted WARS1 turns on TLR2 and TLR4/myeloid differentiation factor-2 (MD-2) signaling to boost innate inflammatory responses (Ahn et al, 2017); the production of cytokines and chemokines such as TNF-α, IL-6, IL-8, and MIP-1α (Lee et al, 2019; Nguyen et al, 2020); and neutrophil infiltration. Patients with sepsis have been found to have high levels of WARS1 in their blood (Ahn et al, 2017). WARS1 has also been reported to be a biomarker for predicting death in sepsis (Choi et al, 2020). In this study, we investigated how WARS1 might be used as a theranostic target for severe sepsis with hypercytokinemia. Further, we highlight the application of WARS1-neutralizing monoclonal antibody, simultaneously suppressing multiple proinflammatory cytokines and chemokines, as a promising therapy for preventing death in systemic severe hyperinflammatory sepsis.

## Results

### Overly secreted WARS1 levels may stratify hypercytokinemic severe sepsis

First, to investigate the potential role of WARS1 as a stratification biomarker for sepsis, a total of 243 participants were enrolled from intensive care units (ICUs) at the Asan Medical Center, and healthy control ($n = 56$), patients diagnosed with either sepsis ($n = 100$) or septic shock ($n = 89$) were included (Fig. EV1A). The 28-day mortality rate was 42%. The baseline characteristics of patients with sepsis and septic shock as well as survivors and non-survivors in 28-day mortality are shown in Tables EV1 and EV2, respectively. Furthermore, we included ICU controls ($n = 30$), which comprised adult patients admitted to the ICU of Seoul National University Hospital following successful resuscitation from non-traumatic cardiac arrest with no infection (Fig. EV1A,B). WARS1 levels were significantly higher in patients with sepsis than in healthy controls ($p < 0.001$, Fig. 1A) or ICU-controls ($p = 0.04$) and in patients with septic shock than in those with sepsis ($p = 0.01$), and WARS1 levels were significantly higher in non-survivors than in survivors ($p = 0.007$, Fig. 1B), indicating a reflection of severity. Additionally, multivariate analysis indicated that levels of WARS1 as well as

lactate and IL-8/CXCL-8 were risk factors associated with 28-day mortality (Table EV3). The optimal cut-off level (106.3 ng/mL) of WARS1 was calculated by Youden's index (Fig. EV1C), and then categorized the patients into two groups (those with high- and low-WARS1). Thus, the probability of survival after 28 days in each group was then compared. The Kaplan–Meier survival analysis showed a significant difference in 28-day mortality between patients with high- and low-WARS1 levels ($p = 0.001$, Fig. 1C). Furthermore, WARS1 levels in the high-WARS1 group showed a moderate positive correlation with TNF-α levels and a low positive correlation with IFN-γ, CXCL8/IL-8, and CCL3/MIP-1α, indicating an association with proinflammatory responses (Fig. 1D). In contrast, the low-WARS1 group showed no correlation with inflammatory cytokines except IFN-γ (Fig. EV2), which is known to induce WARS1 expression (Reano et al, 1993; Rubin et al, 1991). The high-WARS1 group showed a significantly increased absolute neutrophil ($p = 0.002$, Fig. 1E) and monocyte ($p = 0.002$, Fig. 1F) count compared with that in the low-WARS1 group, indicating an increased inflammatory status. Lactate levels in the high-WARS1 group were also higher than those recorded in the low-WARS1 group ($p < 0.001$, Fig. 1G), indicating a more septic shock status. The high-WARS1 group presented a higher SOFA score than the low-WARS1 group, indicating more organ damage ($p < 0.001$, Fig. 1H).

Next, to investigate the relevance of the above human data in a sepsis animal model, we established mild, moderate, and severe sepsis mice by cecum slurry (CS) inoculation intraperitoneally (i.p.) at doses of 16, 18, and 20 mg, and the mortality rates were 23%, 53%, and 90%, respectively, at 72 h (Fig. 2A). Following CS inoculation, plasma WARS1 levels continuously increased and were significantly higher according to severity (Fig. 2B), suggesting its potential as a severity biomarker for sepsis. To further examine the correlation of plasma WARS1 levels with the severity of septic mice, we used mice with moderate sepsis ($CS_{18mg}$), whose mortality rate is similar to that of the human cohort. Plasma WARS1 levels of each mouse were measured at 9 h (significant difference according to severity) and 15 h (peak time), and we found that mice with high-WARS1 levels at 9 h maintained peak WARS1 levels even at 15 h ($p < 0.001$, Fig. 2C), and mice with high WARS1 levels had shorter survival times ($p < 0.001$, Fig. 2D). When mice were split by the median WARS1 levels (17.49 ng/mL) at 15 h, the high WARS1 group showed a significantly poorer survival rate than that of the low-WARS1 group ($p = 0.001$, Fig. 2E). Consistently, the high WARS1 group (at 9 h after CS inoculation) showed significantly higher levels of cytokines and chemokines (Fig. 2F), aspartate aminotransferase (AST), alanine aminotransferase (ALT), and blood urea nitrogen (BUN), indicators of liver and kidney failure, respectively (Fig. 2G–I), than those of the low-WARS1 group, except for blood bacterial colony-forming units (CFU) ($p = 0.09$, Fig. 2J). Taken together, these data suggest that overly elevated WARS1 levels may stratify septic subjects by risk of premature death associated with hypercytokinemia, shock, and organ damage.

### Overly secreted WARS1 is an upstream exacerbating factor for hypercytokinemic severe sepsis

To investigate how WARS1 is involved in sepsis pathogenesis and progression, we first performed an RNA-sequencing (RNA-seq)

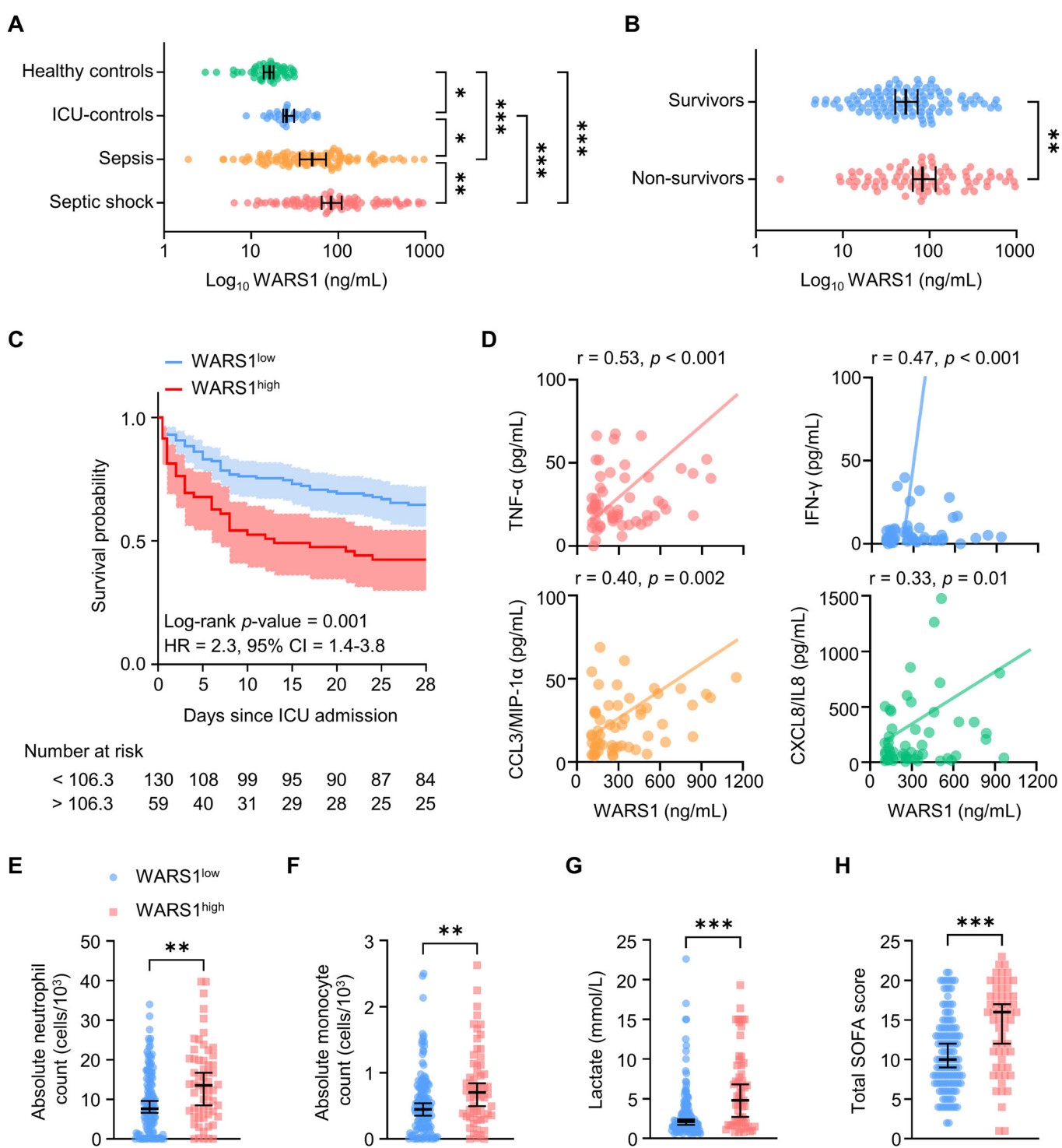

analysis of human peripheral blood mononuclear cells (hPBMCs) treated with recombinant human full-length WARS1 (hFL-WARS1, 50 nM). The absence of endotoxin contamination in hFL-WARS1 was confirmed by the treatment of polymyxin B, which binds to and neutralizes lipopolysaccharide (LPS) (Fig. EV3A). In hPBMCs treated with-hFL-WARS1 vs. control, substantial changes in transcriptome profiles, including 674 differentially expressed genes (DEGs) (414 upregulated, 260 downregulated), were identified

(Fig. 3A). DEGs were defined based on a $|\log_2 \text{ fold-change}| \geq 5$, $p$ value $< 0.01$. Heat maps showed that WARS1 significantly upregulated the expression of pro-inflammatory genes, including interleukin cytokine families (Fig. 3B), chemokine (C-X-C motif) ligand (CXCL), and chemokine (C-C motif) ligand (CCL) families (Fig. 3C). We validated cytokine and chemokine gene expression and production, including IL-6, CXCL8/IL-8, CCL3/MIP-1α, and TNF-α in hPBMCs treated with hFL-WARS1 (Fig. EV3B,C). Next,

◄

**Figure 1.   Elevated WARS1 levels correlate with inflammation, shock, organ damage, and mortality in critically ill patients with sepsis.**

(A) Plasma levels of WARS1 (log10 scale) in healthy controls ($n = 54$), ICU-controls ($n = 30$), patients with sepsis ($n = 100$), and patients with septic shock ($n = 89$). (B) Plasma levels of WARS1 (log10 scale) between survivors ($n = 109$) and non-survivors ($n = 80$). (C) Kaplan–Meier survival plots of the 28-day mortality by optimal cut-off levels of WARS1 (stratified at 106.3 ng/mL). Each line indicates the survival probability over follow-up time with shaded error bands. HR hazard ratio. (D) Correlation between WARS1 levels and cytokine and chemokine levels in the WARS1[high] ($n = 57$) group (stratified above 106.3 ng/mL). Individual correlation results are reported with linear regression lines. (E–H) Absolute neutrophil count (E), absolute monocyte count (F), lactate levels (G), and total SOFA score (H) in the WARS1[low] ($n = 128$–130) vs. WARS1[high] ($n = 58$–59) groups. Data information: Data are presented as median with 95% confidence interval (A, B, E–H). Statistical analysis is performed with Kruskal–Wallis test (A), Mann–Whitney $U$-test (B, E–H), log-rank test (C), and Pearson's correlation coefficient test (D). *$p < 0.05$, **$p < 0.01$, ***$p < 0.001$. Source data are available online for this figure.

the DEGs were classified using gene ontology (GO) analysis and further analyzed using gene set enrichment analysis (GSEA) and ingenuity pathway analysis (IPA). Regarding cellular components and biological processes, in hPBMCs treated with hFL-WARS1, gene expression in the extracellular space and plasma membrane increased the most ($-\log_{10}$ ($p$ value) >5, Fig. 3D), with induction of inflammatory responses and chemotaxis ($-\log_{10}$ ($p$ value) >10, Fig. 3E). GSEA analysis enriched cytokine and chemokine receptor signaling, JAK-STAT signaling, and TLR signaling ($-\log_{10}$ (FDR $q$-value) >2, Fig. 3F). IPA also revealed the most significant changes in pattern recognition receptors (PRRs), triggering receptor expressed on myeloid cells 1 (TREM-1), IL-8 signaling, and TLR signaling ($-\log$ ($p$ value) >1, Fig. 3G). To investigate the transcriptional regulators of gene expression, we performed transcriptional regulatory network analysis using transcriptional regulatory relationships revealed by sentence-based text mining (TRRUST) and a search tool for the retrieval of interacting genes and proteins (STRING). NF-κB transcription factor family members, including *RELB*, *NFKB2*, *REL*, and *IRF1*, predominantly control WARS1-induced gene expression networks (Fig. 3H,I), indicating recapitulating hyperinflammatory sepsis endotypes (Liu and Malik, 2006).

Since the data suggested that WARS1 is an upstream aggravating factor for hyperinflammatory sepsis, we tested whether WARS1 indeed worsens sepsis in vivo. The mildly septic mice (CS$_{16mg}$ inoculation) were i.p. injected with recombinant mouse full-length WARS1 (rWARS1, Fig. 4A). As shown in Appendix Fig. S1, rWARS1 injection decreased the survival rate in a dose-dependent manner, and 20 mg/kg of rWARS1 significantly increased mortality by 40% compared with the CS$_{16mg}$ + PBS group (CS control, Fig. 4B). Injection of rWARS1 into CS$_{16mg}$ mice synergistically increased WARS1 levels in peritoneal lavage fluid (PLF) and plasma compared with the CS control and rWARS1-treated groups, respectively (Fig. 4C,D), and showed dramatically increased cytokine and chemokine levels (Fig. 4E,F). Consistently, the levels of AST, ALT, and BUN (Fig. 4G–I), and blood bacterial CFU were highly increased in the CS$_{16mg}$ + rWARS1 group (Fig. 4J). Taken together, these data suggest that WARS1 is an upstream exacerbating factor and can be used as a therapeutic target for severe sepsis with hypercytokinemia.

## WARS1 neutralization protects severely ill septic mice from lethality

As a therapeutic modality, we created an anti-human WARS1 monoclonal IgG1 antibody (anti-WARS1 MAb) that has subnanomolar affinity for *N*-terminal active domain of human, marmoset, and mouse WARS1 (Fig. EV4) (Ahn et al, 2017). To test the neutralizing activity of the anti-WARS1 MAb, we used marmosets, considering the high similarity between human and non-human

primate (NHP) inflammatory responses (Nelson and Loveday, 2014; 't Hart et al, 2004). Notably, human WARS1 (GenBank: AAH17489.1) shared 94.7% homology with marmoset WARS1 (NCBI reference sequence: XP_035117385.1), which was 89.6% higher than that with murine WARS1 (GenBank: AAH03450.1) (Fig. EV5A). Treatment of human monocytic THP-1 cells with recombinant marmoset WARS1 significantly increased the production of IL-6, CXCL8/IL-8, CCL3/MIP-1α, and TNF-α as observed with human WARS1 (Fig. EV5B). After LPS injection into the marmosets, PBS or anti-WARS1 MAb was administered intravenously (i.v.) (Fig. 5A). A prompt and significant increase in WARS1 levels was observed in the blood of control marmosets after LPS injection, reaching a peak at 4 h (Fig. 5B). Accordingly, increases in IL-6 gene expression started at 0.5 h and reached a peak at 4 h (Fig. 5C), whereas increases in CXCL8/IL-8 and TNF-α gene expression peaked at 0.5 h (Fig. 5D,E), and their production peaked at 4, 2, and 1 h, respectively (Fig. 5F–H). As shown in Fig. 5B, an anti-WARS1 MAb injection drastically neutralized WARS1 levels in the blood. As expected, IL-6 gene expression was significantly suppressed by WARS1 blockage (Fig. 5C), although the effects on protein levels were not significant (Fig. 5F). WARS1 neutralization significantly suppressed CXCL8/IL-8 and TNF-α mRNA expression (Fig. 5D,E) and protein production (Fig. 5G,H). These data suggested that our anti-WARS1 MAb is effective for inhibiting proinflammatory cytokine and chemokine production in vivo.

To evaluate the therapeutic feasibility of targeting WARS1 in sepsis, we tested the effects of the anti-WARS1 MAb on severely ill, septic mice. As presented in Fig. 6A, after PBS (naïve) or CS$_{20mg}$ inoculation, mice were administered either isotype IgG antibody (IgG) or anti-WARS1 MAb i.p at the optimal concentration (Appendix Fig. S2A). In the control IgG group, 90% of the mice died, whereas only 40% of the mice died in the anti-WARS1 MAb group, resulting in a significant difference in survival rate at 72 h ($p < 0.001$, Fig. 6B). We observed that the survival rate was maintained without dead mice for up to 7 days (Appendix Fig. S2B). Corresponding to the decrease in WARS1 in the PLF of septic mice following anti-WARS1 MAb administration (Fig. 6C), cytokine and chemokine production, including IL-6, CXCL2/MIP-2α (human CXCL8/IL-8 homolog), CCL3/MIP-1α, TNF-α, and IFN-γ, was significantly decreased, mainly at 12 h and 15 h, as compared with the IgG group (Fig. 6D). Similarly, in the plasma of septic mice, WARS1, cytokine, and chemokine levels were decreased in the anti-WARS1 MAb-treated group compared with the IgG group (Fig. 6E,F). Subsequently, the degree of organ damage was examined. The IgG group showed severe alveolar destruction and inflammatory cell infiltration, as shown by histological staining (Fig. 7A). Immunofluorescence images also showed an increase in Ly6G-positive leukocytes

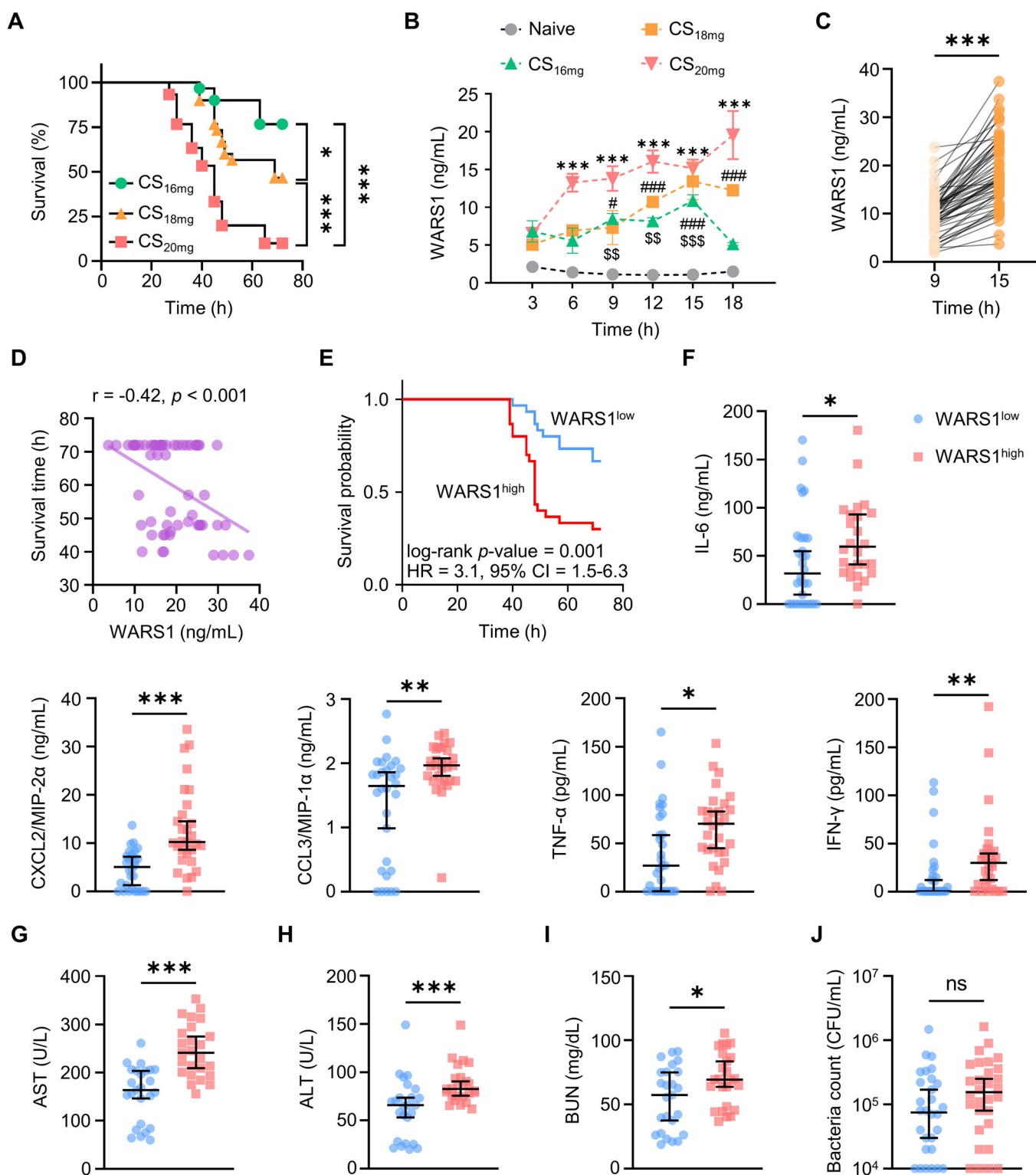

(mainly neutrophils and monocytes) infiltrating the lung (Fig. 7B), as well as myeloperoxidase (MPO) levels and MPO activity in control IgG mice (Fig. 7B,C). Furthermore, significantly increased *CCL2/MCP-1*, *CCL7/MCP-3*, *CXCL1/KC*, *CXCL5/ENA-78*, and *CXCL13/BLC* expression was observed (Fig. 7D). However, the anti-WARS1 MAb-treated group did not exhibit alveolar disruption and showed

decreased MPO levels, suppressed neutrophil infiltration, and chemokine gene expression (Fig. 7A–C). Likewise, hepatic hemorrhage, observable damage to kidney parenchyma, vacuolar degeneration, and interstitial edema were significantly decreased in the anti-WARS1 MAb group (Fig. 7E,F). ALT, AST, and BUN levels were significantly decreased compared with those in the IgG group

**Figure 2. WARS1<sup>high</sup> levels are hallmarks of hyperinflammation, organ damage, and mortality in sepsis mouse model.**

(A) Kaplan–Meier survival plots for mice inoculated with CS at different doses of 16–20 mg ($n = 30$ per group). (B) Release kinetics of plasma WARS1 in sepsis mice with CS$_{16–20mg}$ ($n = 2$–4 per group). Data are presented as mean ± SEM. Naïve vs. CS$_{16mg}$, $^{\$\$}p < 0.01$, $^{\$\$\$}p < 0.001$; Naïve vs. CS$_{18mg}$, $^{\#}p < 0.05$, $^{\#\#\#}p < 0.001$; Naïve vs. CS$_{20mg}$, $^{***}p < 0.001$. (C) Plasma WARS1 levels at 9 and 15 h after CS$_{18mg}$ inoculation ($n = 60$). Each line indicates a connection of individual symbols between the two times. (D) Correlation between WARS1 levels and survival time in CS$_{18mg}$-inoculated mice ($n = 60$). Individual correlation results are reported with linear regression lines. (E) Kaplan–Meier survival plots of WARS1<sup>low</sup> vs. WARS1<sup>high</sup> groups (stratified at 17.49 ng/mL) 15 h after CS$_{18mg}$ inoculation ($n = 30$ per group). (F–J) Levels of cytokine and chemokine (F), AST (G), ALT (H), BUN (I), and blood bacterial CFU (J) at 9 h after CS inoculation in the WARS1<sup>low</sup> vs. WARS1<sup>high</sup> groups ($n = 25$–30 per group). Data information: Data are presented as median with a 95% confidence interval (F–J). Statistical analysis is performed with log-rank test (A, E), ANOVA with Bonferroni corrections (B), Mann–Whitney $U$-test (C, F–J), and Pearson's correlation coefficient test (D). ns, not significant; $*p < 0.05$, $**p < 0.01$, $***p < 0.001$. Source data are available online for this figure.

(Fig. 7G–I), whereas blood bacterial CFU showed only a tendency to decrease in the anti-WARS1 MAb group ($p = 0.06$, Fig. 7J).

Finally, to mimic human clinical scenarios, we injected antibiotics (gentamicin (Gen), 3 mg/kg) into the mice 2 h after CS$_{20mg}$ inoculation. The mice were then administered with IgG or anti-WARS1 MAb (10 mg/kg, Fig. 8A). In the IgG control group, 90% of the mice died within 72 h after CS inoculation, whereas in the IgG + Gen group and anti-WARS1 MAb + Gen group, 50% and 10% of the mice died, resulting in 50% ($p = 0.04$) and 90% ($p < 0.001$) survival rates, respectively, compared with the IgG control group (Fig. 8B). WARS1 was successfully neutralized in the PLF and plasma of the anti-WARS1 MAb + Gen group (Fig. 8C,D), leading to a significant decrease in MPO activity, BUN, and creatinine levels in comparison with those in the IgG group, whereas IgG + Gen only showed a limited effect (Fig. 8E–G). As shown in Fig. 8H,I, although IgG + Gen treatment significantly decreased cytokine and chemokine production in the PLF and plasma compared with the IgG group, the addition of anti-WARS1 MAb further normalized production levels to those of the naïve groups. Taken together, these results suggest that WARS1 neutralization and its combination with antibiotics protects severely ill septic mice from lethality by dampening excessive cytokine and chemokine productions and organ destruction in severe sepsis.

## Discussion

Given the variability and difficulties of classifying patients with sepsis in clinical settings, the use of biomarker panels during sepsis pathogenesis is a step forward toward the identification of distinct populations that would allow more focused clinical trials. Our study suggested the possibility that WARS1-guided patient stratification and targeted therapy may be applicable for critically ill sepsis patients with hypercytokinemia. In our cohort analysis, critically ill septic patients with high WARS1 levels in their blood had higher lactate levels, a higher SOFA score, and a higher risk of premature mortality. In addition, high TNF-α, IFN-γ, CCL3/MIP-1, and CXCL8/IL-8 levels, along with an increase in both absolute neutrophil and monocyte count, were consistent with the high secretion of WARS1. The correlation of WARS1 levels and their clinical relevance in a CS mouse model reproduced the human data, where high WARS1 secretion had a reliable ability to predict severity, organ dysfunction, and early death associated with hypercytokinemia. These findings provide an indication that by evaluating high levels of plasma WARS1 upon ICU admission, sepsis patients who would benefit from an anti-WARS1 MAb injection may be identified and administered the therapeutic at the appropriate stage of sepsis. Nonetheless,

considering the pathological complexity of sepsis, we do not think that measuring a specific WARS1 level at a given point would work as a sole determinant to stratify the patients who would respond to the anti-WARS1 MAb treatment. Systemic research should be followed involving the prospective analysis of multiple cohorts. For instance, how plasma WARS1 levels change over time in relation to the immunopathological state, disease severity, clinical parameters, biomarkers, and recovery or death would probably provide insights into the precise theranostic use of anti-WARS1 MAb. Further, since vertebrate WARS1 shares a high degree of similarity in its amino acid sequence, other models, such as the minipig or nonhuman primate sepsis model, could probably be used to simulate human sepsis clinical conditions more accurately.

Our investigation revealed that excessive secretion of WARS1 represents a potential therapeutic target in severe hypercytokinemic sepsis. This condition, which leads to tissue parenchymal destruction and organ failure, is caused by a breakdown of disease tolerance that is not directly associated with immune resistance (i.e., pathogen load) (Soares et al, 2017). This may explain why antimicrobial approaches are ineffective in treating diseases such as sepsis. Consequently, disease tolerance has been recognized as a reasonable pharmaceutical objective to control host stress and tissue damage, irrespective of the bacterial burden (Soares et al, 2017). Injecting WARS1 protein into mice with mild sepsis resulted in substantial tissue damage, organ failure, and death, accompanied by the substantial production of proinflammatory cytokines and chemokines, which highlights the direct involvement of WARS1 in the pathophysiology of sepsis and its progression to severity. Anti-WARS1 MAb therapy, a novel pharmacological intervention, did not directly alter the bacterial load (Fig. 7J); however, it is thought to have preserved disease tolerance via its ability to inhibit the expression of multiple cytokines and chemokines at the infection site, thereby minimizing parenchymal cell damage (Chen et al, 2018). The antibody treatment exerted anti-inflammatory effects via simultaneous inhibition of IL-6, CXCL8/IL-8, CCL3/MIP-1, TNF-α, and IFN-γ production, and had direct and considerable suppressive effects on excessive CCL and CXCL gene expression associated with sepsis immunopathogenesis (Mei et al, 2010; Mercer et al, 2014; Paudel et al, 2019; Shalova et al, 2015; Souto et al, 2011). The antibody effects were seen in WARS1-induced gene profiling mirrors of hyperinflammatory sepsis endotypes, such as Molecular Diagnosis and Risk Stratification of Sepsis (MARS) 2, inflammopathic endotype, and delta-inflammatory status phenotype, which are characterized by the upregulation of genes implicated in cytokine and chemokine signaling, TLRs, and NF-κB activation (Scicluna et al, 2017; Seymour et al, 2017; Sweeney et al, 2018). Finally, WARS1 neutralization could improve survival and, in conjunction with antibiotics, almost completely save the lives of mice from severe sepsis.

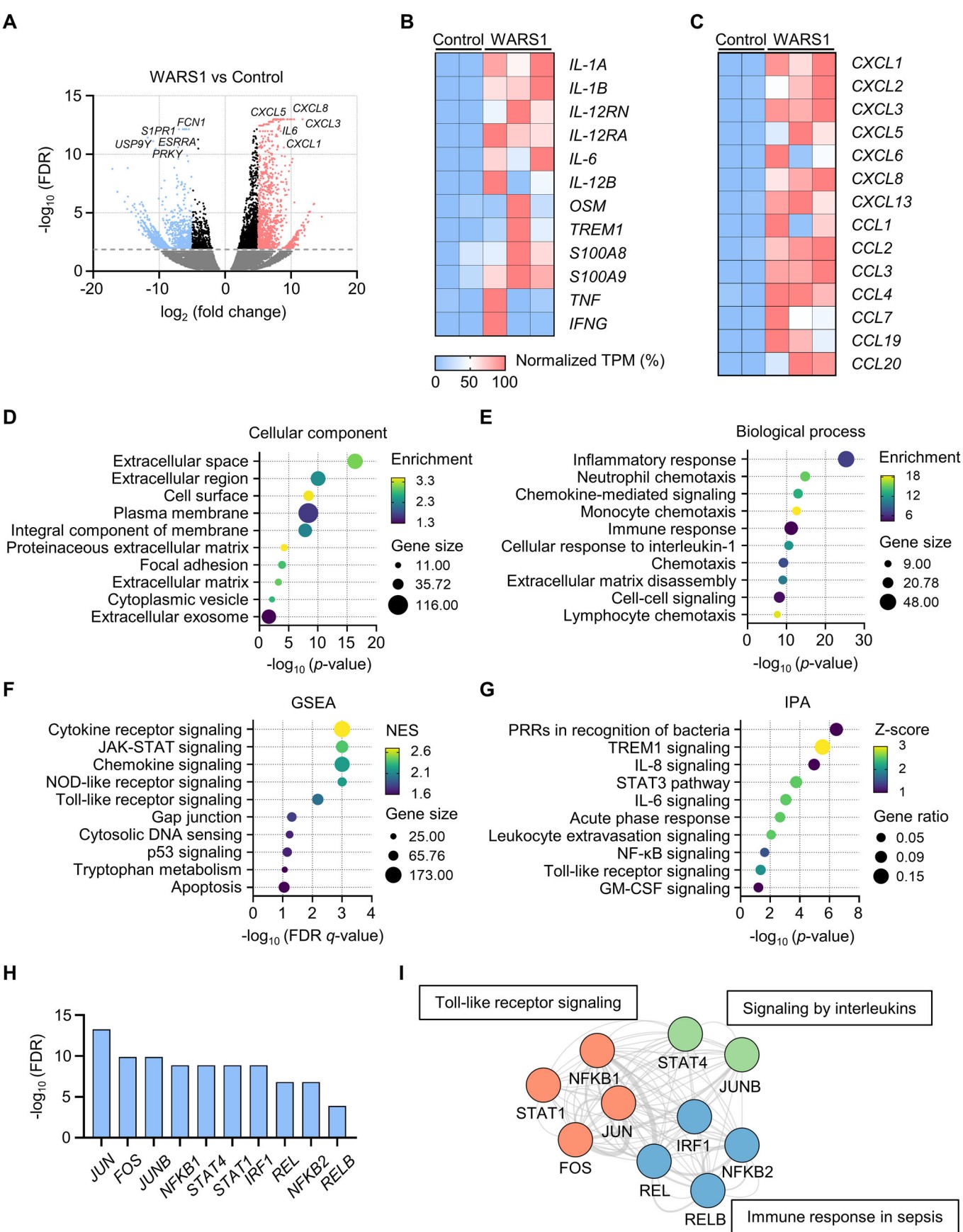

**Figure 3.  WARS1 activated transcriptomic profiling designates hyperinflammatory sepsis gene expression in hPBMCs.**

(A) Volcano plot depicting significantly up-regulated (red) or down-regulated (blue) genes (|log$_2$ fold change| ≥ 5, false discovery rate (FDR) <0.01) from hFL-WARS1 (50 nM) treated hPBMCs compared with controls. |log$_2$ fold change| ≤ 5 genes are shown in black, and FDR >0.01 genes fall under the gray dashed line. (B, C) Heat map showing selected upregulated genes in hPBMCs treated with hFL-WARS1 compared with controls, divided into cytokines (B) and chemokines (C), respectively. TPM, transcripts per million. (D, E) Functional annotations of cellular components (D) and biological processes (E) from hFL-WARS1-derived upregulated genes using database for annotation, visualization and integrated discovery (DAVID)-GO analysis tool. (F, G) Signaling pathway of GSEA (F) and IPA (G) from hFL-WARS1-derived upregulated genes. NES, normalized enrichment score. (H) Upstream transcription factors activated by hFL-WARS1 using TRRUST analysis tool. (I) Signaling pathway based on transcription factors activated by hFL-WARS1 using STRING analysis tool. Data information: The color and size of the dots represent the scale score and expression ratio of each value, respectively (A, D–G). Statistical analysis is performed with Student's t test (A, F), and Fisher's exact test (D, E, G). Source data are available online for this figure.

As a therapeutic modality, the anti-WARS1 MAb seems promising as, in comparison to a single cytokine inhibitor, it concurrently suppresses multiple cytokines and chemokines and a range of pathways that contribute to the development and progression of the disease. Furthermore, eliminating blood-circulating WARS1 (a transiently released endogenous TLR2 and TLR4 ligand) could have minimal impact on TLR receptors, which are situated on diverse cells and regulate complex signaling, thus being considered safer than TLR4 antagonism (Kawasaki and Kawai, 2014).

In the current medical situation of sepsis, where mortality is unacceptably high in the absence of a reliable biomarker and specifically established therapy, our data imply a meaningful clue that a blood-circulating WARS1-targeting theranostic approach may help to improve the management of critically ill patients with sepsis based on a precision medicine strategy.

## Methods

### Human study

This study involved the analysis of prospective cohorts of adult patients with sepsis ($n = 100$) or septic shock ($n = 89$) and healthy controls ($n = 54$) from Asan Medical Center. Adult (older than 19 years old), critically ill patients admitted to the medical intensive care unit (ICU) of each participating hospital were eligible, and patients diagnosed with sepsis or septic shock according to the Third International Consensus Definitions for Sepsis and Septic Shock (Sepsis-3) were included in the study (Singer et al, 2016). Written informed consent was obtained from patients or their legally authorized representative prior to enrollment, and the experiment conformed to the principles set out in the WMA Declaration of Helsinki and the Department of Health and Human Services Belmont Report. The study was approved by the Institutional Review Boards of Asan Medical Center (No. 2016-0186). Baseline demographics, clinical details, including the SOFA score collected in the first 24 h after admission to the ICU, laboratory data, and relevant outcomes were recorded. 19 mL of whole blood samples were collected from all patients within 24 h of ICU admission. Serum was separated and stored at −80 °C until analysis. Sampling protocols are available from a previously published study (Ma et al, 2018).

The prospective biobank of post-cardiac arrest patients was analyzed as part of the ICU control group. The Institutional Review Board approved the prospective patient enrollment and sample preservation (SNUH IRB No. 1707-012-865) as well as this analysis (SNUH IRB No. 2307-149-1452). The study protocol was registered

at ClinicalTrials.gov (NCT01670383). Adult patients admitted to the ICU of the study hospital following successful resuscitation from nontraumatic cardiac arrest were prospectively screened. Blood samples were collected at admission and 24 h after admission after obtaining written informed consent from the patient or next of kin. Serum was separated and stored at −80 °C until analysis. A total of 30 patients enrolled between March 2013 and December 2014 were analyzed in this study.

### Animal experiments

Murine animal experiments were complied with the Association for Assessment and Accreditation of Laboratory Animal Care International guidelines and approved (LCDI-2020-0088) by the Lee Gil Ya Cancer and Diabetes Institute. Wild-type C57BL/6 J mice (both males and females, 9–11 weeks old) were purchased from Daehan Biolink Co., Ltd. All mice were housed in an air conditioner that was set to operate at a temperature of 22.7 °C, a humidity of 48.8 to 63%, and a differential pressure of −2.24 mmAq. CS stock was prepared according to a previously reported method (Starr et al, 2014). Mice were i.p. inoculated with CS stock (16–20 mg per mouse) according to experiment purpose, and mice were i.p. injected 4 h after CS inoculation with a single dose of rWARS1 (5–20 mg/kg) or an equivalent volume of PBS. For anti-WARS1 MAb administration, mice were i.p. injected three times (4, 8, and 12 h) after CS inoculation with anti-WARS1 MAb (a total of 2.5–10 mg/kg) or an equivalent amount of the isotype control IgG (Bio X cell, cat. BP0301). For antibiotic administration, the mice were subcutaneously injected with gentamicin (Sigma-Aldrich, G1914, 3 mg/kg). Blood was collected by cardiac puncture method at each time point in a sodium heparin anticoagulated tube (BD Vacutainer, cat. 367871) or a serum-separating tube (BD Vacutainer, cat. 367955) and centrifuged at 4000 rpm at RT for 15 min to isolate plasma and serum, respectively. For the PLF, 3 mL of PBS is injected into the peritoneum of the mice, followed by a massage for 2–3 min, collected, and centrifuged at 1500 rpm at RT for 5 min. The supernatants were stored at −80 °C for further analysis. All experiments were conducted using age- and sex-matched mice. Samples from each animal were prepared in duplicates or triplicates for measurements, and the survival experiments were repeated two or three times.

The common marmoset (*Callithrix jacchus*) study complied with the Association for Assessment and Accreditation of Laboratory Animal Care International guidelines and was approved (KBIO-IACUC-2020-216; KBIO-IBC-2021-014) by the Institutional Animal Care and Use Committee and the Institutional Biosafety Committee of the Laboratory Animal Center of the Osong Medical Innovation Foundation. Both males and females at 2-4 years old were used. Marmosets were i.v. injected with LPS (Sigma

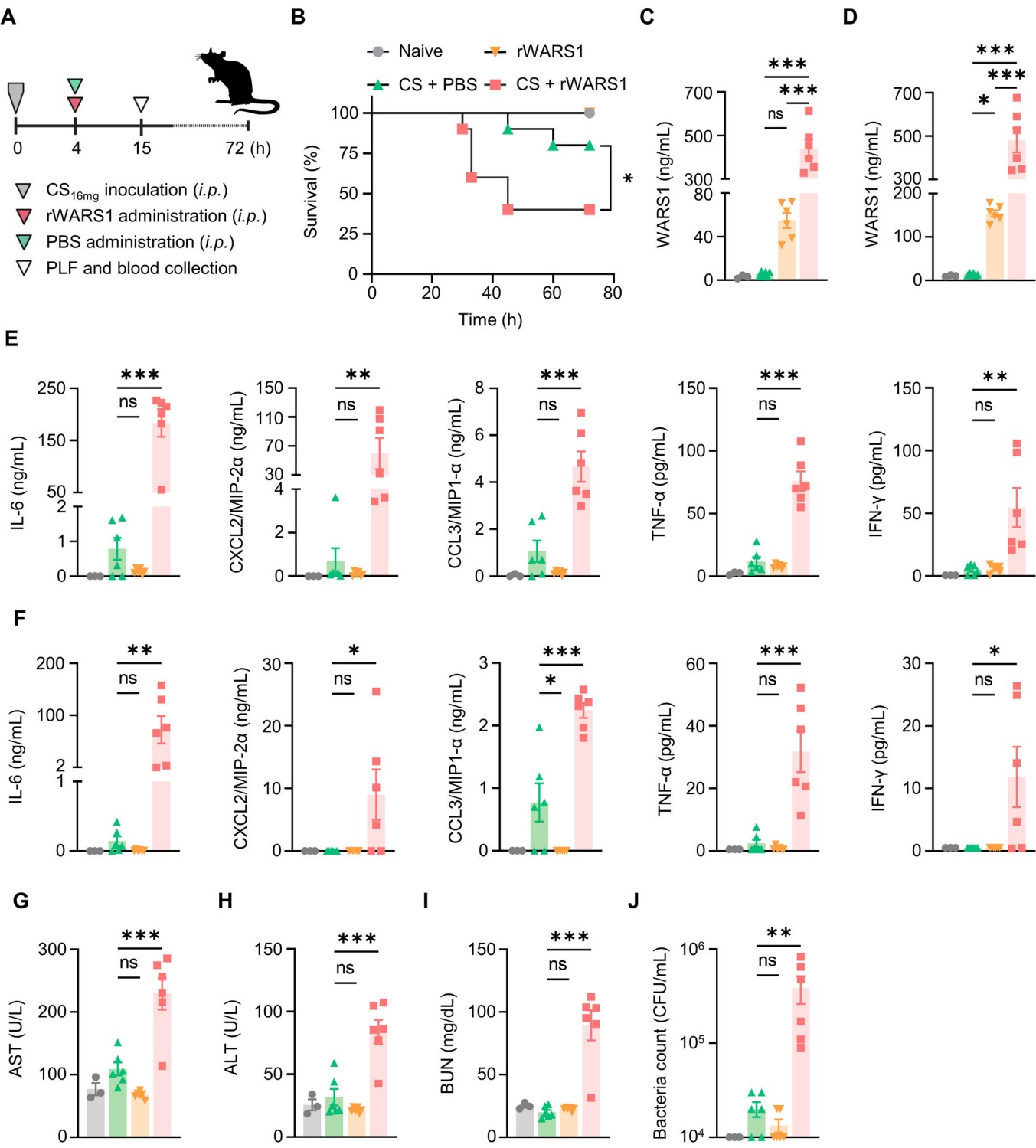

**Figure 4. WARS1 administration exacerbates cytokine and chemokine production, organ damages, and mortality in septic mice.**

(A) Experimental scheme of CS$_{16mg}$ inoculated mild septic mouse model. (B) Kaplan–Meier survival plot for mice administered with PBS or rWARS1 (20 mg/kg) after CS$_{16mg}$ inoculation, naive mice were administered with PBS ($n = 10$ per group). (C, D) WARS1 levels in PLF (C) and plasma (D) at 15 h after CS inoculation ($n = 3$–6 per group). (E, F) Levels of cytokine and chemokine in PLF (E) and plasma (F) at 15 h after CS inoculation ($n = 3$–6 per group). (G–J) Serum levels of AST (G), ALT (H), BUN (I), and blood bacterial CFU (J) at 15 h after CS inoculation ($n = 3$–6 per group). Data information: Data are presented as mean ± SEM (C–J). Statistical analysis is performed with log-rank test (B), and ANOVA with Bonferroni corrections (C–J). ns, not significant; *$p < 0.05$, **$p < 0.01$, ***$p < 0.001$. Source data are available online for this figure.

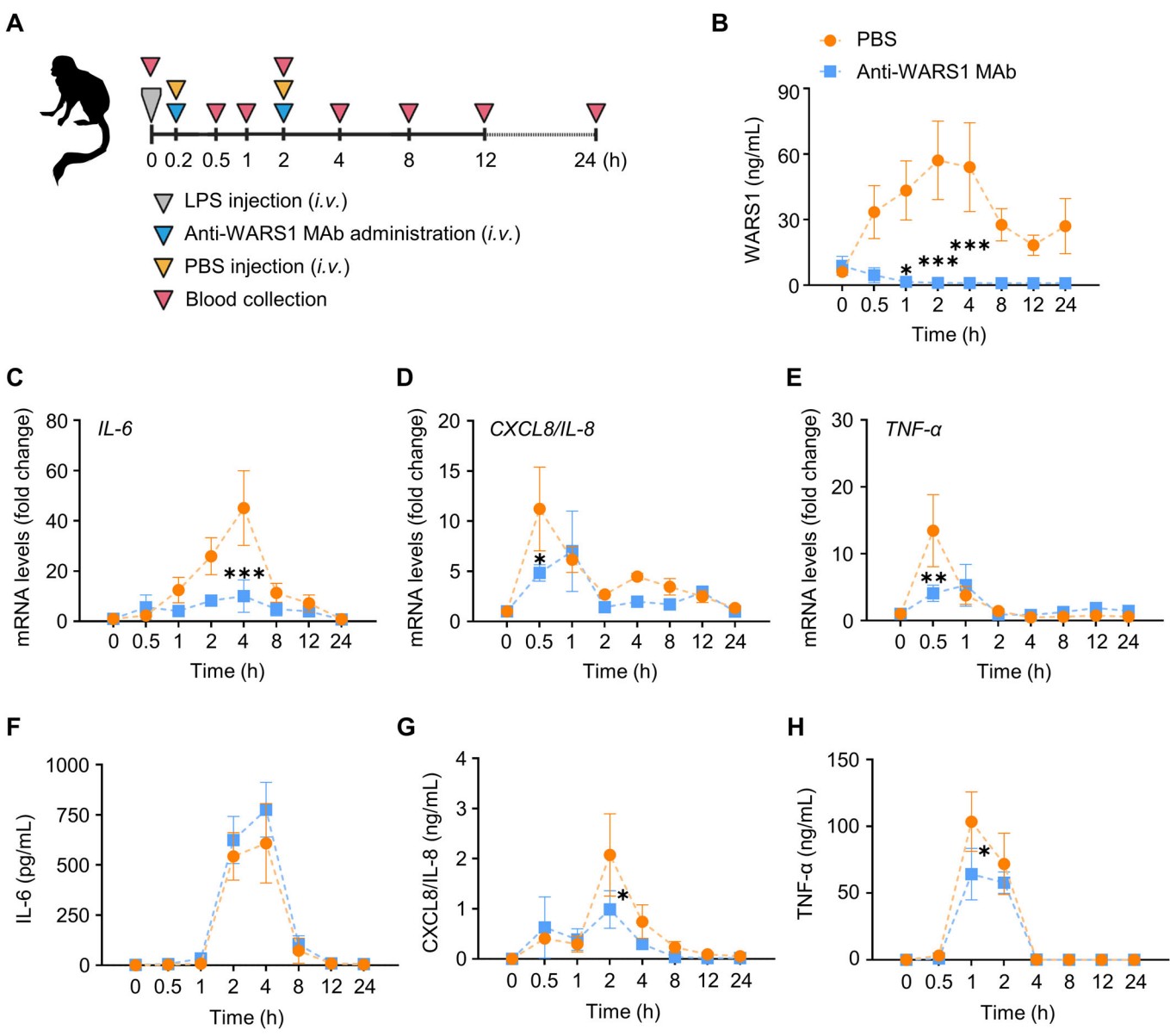

**Figure 5.** Anti-WARS1 monoclonal antibody administration suppresses proinflammatory cytokine gene expressions in LPS-injected marmosets.

(A) Experimental scheme of the LPS (100 μg/kg) injected marmoset endotoxemia model. (B) Plasma WARS1 levels at indicated time points after LPS injection ($n = 5$ per group). (C–E) Gene expression of cytokine and chemokine in whole blood of marmosets at indicated time points after LPS injection ($n = 4$–5 per group). (F–H) Plasma levels of cytokine and chemokine at indicated time points after LPS injection ($n = 3$–5 per group). Data information: Data are presented as mean ± SEM. Statistical analysis is performed with ANOVA with Bonferroni corrections (B–H). *$p < 0.05$, **$p < 0.01$, ***$p < 0.001$. Source data are available online for this figure.

Aldrich, *E. coli* O111:B4, 100 μg/kg per head). For anti-WARS1 MAb administration, marmosets were i.v. injected twice (at 0.2 and 2 h) after LPS injection with anti-WARS1 MAb (a total of 5 mg/kg) or an equivalent volume of PBS. Investigators were blinded to animal experimental procedures, including treatment, sample collection, and measurements.

**Enzyme-linked immunosorbent assay (ELISA)**

WARS1 levels in the plasma of patients with sepsis, CS-inoculated mice, and LPS-injected marmosets were determined according to the following protocols: Immunoplates were coated with anti-

WARS1 at 4 °C overnight and blocked at 25 °C (RT) for 1 h; samples were incubated at RT for 1 h with a WARS1 detection antibody (Abcam, ab228724, 1:500) and a secondary antibody, HRP-conjugated anti-rabbit IgG (Cell Signaling, 7074S, 1:3000), at RT for 1 h; 3,3′,5,5′-tetramethylbenzidine (TMB) substrate (BD, cat. 555214) was added and incubated at RT for 10 min; then, sulfuric acid solution (Samchun, cat. S2129) was added to terminate the reaction.

CRP (R&D System, DY1707), PCT (RayBiotech, cat. ELH-PROCALC), CXCL8/IL8, CCL3/MIP-1α, TNF-α, and IFN-γ (Milliplex Map Kit Human High Sensitivity T cell Magnetic Bead Panel, EMD Millipore) levels in the plasma of patients with sepsis,

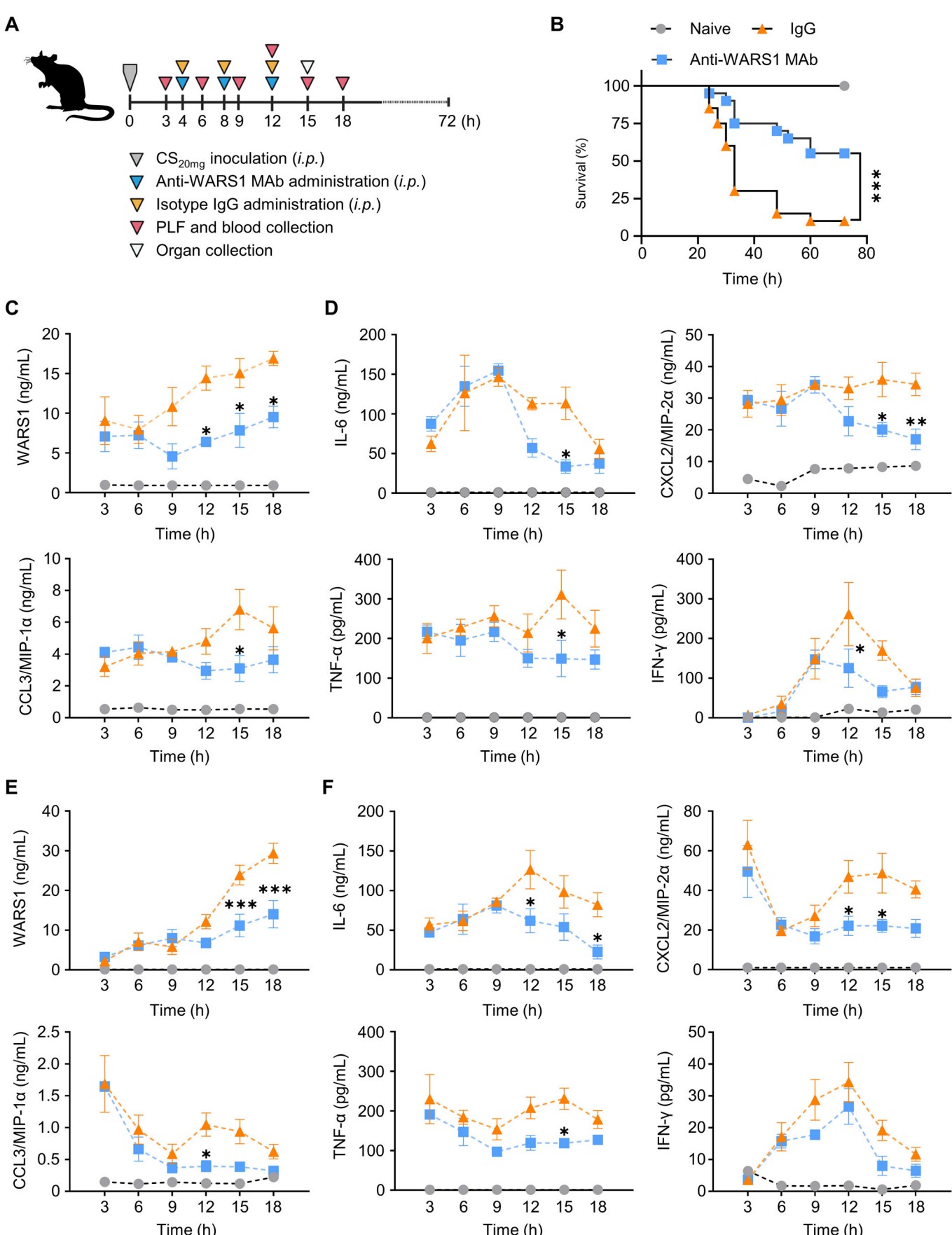

**Figure 6. WARS1 neutralization suppresses excessive inflammation and premature deaths in severely ill septic mice.**

(A) Experimental scheme for $CS_{20mg}$ inoculated severe septic mouse model. (B) Kaplan–Meier survival plot for mice administered with control IgG or monoclonal anti-WARS1 antibody (10 mg/kg) after $CS_{20mg}$ inoculation, naïve mice were administered with PBS ($n = 20$ per group). The survival rate experiments to determine the efficacy of anti-WARS1 MAb were repeated two to three times. (C, D) Levels of WARS1 (C), cytokine and chemokine (D) in PLF at the indicated time points ($n = 5$–6 per group). (E, F) Levels of WARS1 (E), cytokine and chemokine (F) in plasma at indicated time points ($n = 3$–6 per group). Data information: Data are presented as mean ± SEM. Statistical analysis is performed with log-rank test (B), and ANOVA with Bonferroni corrections (C–F). *$p < 0.05$, **$p < 0.01$, ***$p < 0.001$. Source data are available online for this figure.

and IL-6 (R&D System, DY206), CXCL8/IL-8 (R&D System, DY208), CCL3/MIP-1α (R&D System, DY270), and TNF-α (R&D System, DY210) levels in the culture supernatant of hPBMCs, and IL-6 (BioLegend, cat. 431304), CXCL2/MIP-2α (R&D System, DY452), CCL3/MIP-1α (R&D System, DY450), TNF-α (BioLegend, cat. 430904), and IFN-γ (BioLegend, cat. 430804) levels in the PLF and plasma of CS-inoculated mice, and IL-6 (U-CyTech, cat. CT346A), CXCL8/IL-8 (Biolegend, cat. 431504), and TNF-α (U-CyTech, cat. CT342A) levels in plasma of LPS-injected marmosets were quantified using the above commercially available ELISA kits, according to the manufacturer's protocols. The absorbance of the sample was measured at 450 nm using a Versa Max microplate reader (Molecular devices) and analyzed using Soft max pro-5, and fluorescence was measured using the Bio-Plex 200 system (Bio-Rad) and analyzed using Bio-Plex Manager software.

### Cell culture

Blood was obtained from healthy volunteers, and the study protocol was approved by the Gachon University Bioethics Committee (IRB approval number: GCIRB2017-304). hPBMCs were isolated using a cell preparation tube (Becton Dickinson, cat. 362761) according to the manufacturer's protocol. hPBMCs were grown in Roswell Park Memorial Institute (RPMI-1640) medium supplemented with 10% fetal bovine serum (FBS), 1% penicillin and streptomycin (Pen-strep, Invitrogen, cat. 15070063).

THP-1 (human monocytic cell line, ATCC TIB-202) and J774A.1 (murine macrophage cell line, ATCC TIB-67) cells were purchased from the American Type Culture Collection. THP-1 cell lines were grown in RPMI-1640 medium supplemented with 10% FBS, 1% pen–strep, and 50 μM β-mercaptoethanol. Upon differentiation, THP-1 cells were treated with phorbol-12-myristate-13-acetate (PMA, Sigma Aldrich, P1585) at 50 ng/mL for 3 days and recovered for 24 h. J774A.1 cell lines were grown in Dulbecco's Modified Eagle Medium (DMEM, Welgene) supplemented with 10% FBS and 1% Pen-strep.

### Library construction and RNA-seq

To obtain high-throughput human transcriptome data, we implemented Illumina-based NGS sequencing. Total RNA was extracted from individual samples and controls using Trizol reagent (Invitrogen) according to the manufacturer's protocol. Total RNA was then quantitated using a Nanodrop spectrophotometer (Thermo Scientific) and quality assessed using the RNA 6000 Nano assay kit (Agilent) and Bioanalyser 2100 (Agilent). NGS sequencing libraries were generated from 1 μg of total RNA using a TruSeq RNA Sample Prep Kit (Illumina) according to the manufacturer's protocol. In brief, the poly-A-containing RNA molecules were purified using the poly-T oligo-attached magnetic

beads. After purification, the total polyA+RNA was fragmented into small pieces using divalent cations. The cleaved mRNA fragments were reverse transcribed into first-strand cDNA using random primers. Short fragments were purified with a QiaQuick PCR extraction kit and resolved with EB buffer for end reparation and addition of poly (A). Subsequently, the short fragments were connected with sequencing adapters. Each library was separated by adjoining distinct MID tags. The resulting cDNA libraries were then paired-end sequenced ($2 \times 101$ bp) for samples with the Novaseq™ 6000 system (Illumina).

### Preprocessing and expression estimation

Paired end sequence files from eleven samples (Fastq: R1, R2) were obtained and processed using Trimmomatic-0.36 (Bolger et al, 2014) with the following parameter settings: leading: 5, trailing: 5, sliding window: 4:15, and minlen: 36. After quality scoring and assessing the read lengths, RNA-Seq reads were mapped to the human reference genome GRCh38 (Kersey et al, 2018) (Gencode release 12) using STAR (Dobin et al, 2013) with default parameters. Accurately quantifying the expression level of a gene from RNASeq reads was achieved using RSEM (Li and Dewey, 2011), which assembles individual transcripts from RNA-Seq reads that have been aligned to the genome sequences. Next, TPM was calculated for each transcribed fragment in the sample to quantify the expression level. To compare each sample, TPM underwent global normalization and was used for further analysis. Principal component analysis (PCA) was applied to the obtained gene expression profiles using the R package stats (R Core Team, 2013).

### DEGs analysis and IPA

The normalized expression profiles of DEGs expressing more than 0.3 TPM and five read counts were used for DEGs analysis using EdgeR (Robinson et al, 33). The expression profile of each gene was scaled to a z-score and hierarchically clustered using a complete linkage method. Visualization was achieved using the ggplot2 library in R packages. Expression profiles were then classified into clusters of similar patterns, and enriched pathways, networks, and functions were analyzed using GSEA (Subramanian et al, 2005) and IPA (QIAGEN Inc.) (Kramer et al, 2014) for each cluster. The significance of each pathway and function was assessed using a significance of $p < 0.05$, and a cutoff of three genes per pathway. The binary heatmap was then constructed to reveal all genes involved in the significant pathways using an in-house R script.

### Quantitative real-time (qRT)-PCR

cDNA was synthesized by SimpliAmp the Thermal Cycler (Applied Biosystem) according to the manufacturer's instructions using the

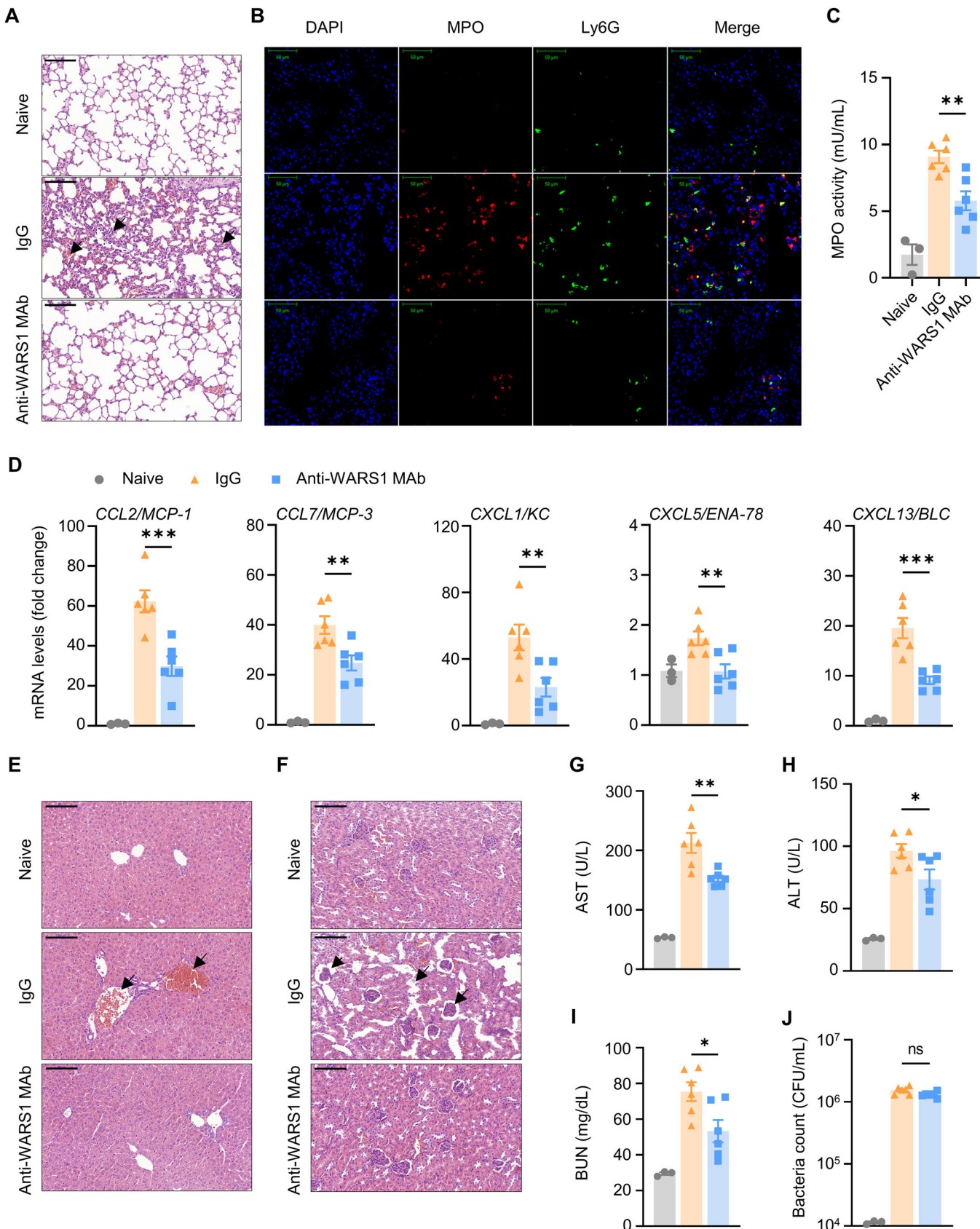

**Figure 7. WARS1 neutralization prevents organ damages in severely ill septic mice.**

(A) Representative images of H&E-stained lung sections at 15 h after CS inoculation (magnification, 200×; scale bars, 100 μm). Arrows indicate areas of inflammation. (B) Representative images of IF-stained lung sections at 15 h after CS inoculation. Nuclei stained with DAPI (blue); MPO with Alexa Fluor 647 (red); neutrophils with Ly6G and Alexa Fluor 488 (green), (magnification, ×200; scale bars, 50 μm). (C) MPO activity in lung homogenates at 15 h after CS inoculation ($n = 6$ per group). (D) Gene expression of CCL and CXCL chemokine in lung homogenates at 15 h after CS inoculation ($n = 6$ per group). (E, F) Representative images of H&E-stained liver (E) and kidney (F) sections at 15 h after CS inoculation (magnification, ×200; scale bars, 100 μm). Arrows indicate areas of inflammation. (G–J) Serum levels of AST (G), ALT (H), BUN (I), and blood bacterial CFU (J) at 15 h after CS inoculation ($n = 6$ per group). Data information: Data are presented as mean ± SEM. Statistical analysis is performed with ANOVA with Bonferroni corrections (C, D, G–J). ns, not significant; $^*p < 0.05$, $^{**}p < 0.01$, $^{***}p < 0.001$. Source data are available online for this figure.

PrimeScript RT reagent kit (Takara, cat. RR037A). Quantitative real-time (qRT)-PCR was performed in a 10 μL reaction containing 1 μL of synthesized cDNA, 5 μL of SYBR green PCR master mix (Applied Biosystem, 4367659), 1 μL of primers, and 3 μL of diethylpyrocarbonate (DEPC)-treated water. Analysis was performed according to the instructions in the 7300 real-time PCR system (Applied Biosystem) and the Bio-Rad CFX 384 real-time system (Bio-Rad). Glycerinaldehyde-3 phosphate-dehydrogenase (GAPDH) was used as an endogenous housekeeping control gene and calculated as relative expression (ΔCT). Primer sequences are listed in Table EV4.

## Recombinant protein purification

The human, marmoset, and mouse full-length WARS1 sequences were cloned in the pET-28a (+) (His-tag) vector, and recombinant proteins were overexpressed in the *E. coli* Rosetta 2 (DE3) strain. The bacterial cells were lysed by sonication (Sonics, 2 s duration with a 90 s interval). The His-tagged proteins in suspension were repeatedly passed through the Ni-NTA chromatography column (Bio-Rad, cat. 7371512). The purified proteins were filtered using a centrifugal filter tube (Millipore, UFC903024 and UFC810024), and endotoxin was removed via a spin column (Thermo Scientific, 88277). Endotoxin levels were measured using the LAL chromogenic endotoxin quantification kit according to the manufacturer's protocol (Thermo Scientific, 88282).

## Histological examination

Organs from sacrificed mice were fixed in 4% formaldehyde solution (Duksan, UN2209) at 4 °C overnight. Specimens of 4% formalin-fixed tissue were embedded in paraffin using an auto processor and subsequently sectioned at 9 μm. Hematoxylin and eosin (H&E) staining was performed according to the routine protocol, and slides were scanned using the Pannoramic Scan II (3DHistech). Histological examination was performed in a blind manner by requesting a histopathologist from the Center of Animal Care and Use of Lee Gil Ya Cancer and Diabetes Institute.

## Immunofluorescence (IF) staining and confocal microscopy

Lungs from sacrificed mice were stored at −80 °C. Frozen embedded tissues were sectioned at 9 μm thick. Slides were incubated at 4 °C overnight with the primary antibodies: myeloperoxidase (Abcam, ab208670, 1:100); Ly6g (Abcam, ab25377, 1:200); and applied at RT for 1 h with the secondary antibodies: Alexa Fluor 647 (Abcam, ab150079,

1:1000); and Alexa Fluor 488 (Abcam, ab150157, 1:1000). DAPI solution (Abcam, ab104139) was applied, and images were acquired via LSM-700 (Zeiss) confocal microscopy.

## MPO activity

Levels of MPO activity were determined using the MPO colorimetric activity assay kit (Sigam-Aldrich, MAK068) according to the manufacturer's instructions. Lungs were homogenized in MPO assay buffer. The standard curve was plotted using the TNB standard; changes between the blank and each sample were determined using $\Delta A412 = (A412)$ sample blank $-$ (A412) sample. Absorbance changes as the TNB reagent/sample is consumed by taurine chloramine in MPO production. The MPO activity of a sample was then determined using the following equation:

$$\text{MPO activity} = \frac{B \times \text{Sample dilution factor}}{(\text{Reaction time}) \times V}$$

where $B$ is the amount (nmole) of TNB consumed; reaction time is in minutes, at the point when stop mix was added; $V$ is the sample volume (mL); and MPO activity is reported as nmole/min/mL = milliunit/mL. One unit of MPO activity is defined as the amount of enzyme that hydrolyzes the substrate and generates taurine chloramine to consume 1.0 μM of TNB per minute at RT.

## Measurement of serum parameters

Sera from sacrificed mice were isolated, and AST, ALT, BUN, and creatinine levels were quantified using the blood biochemical analyzer (7180 Hitachi). Serum was injected into a vacutainer containing clot activator and coagulated at RT for 15 min. The coagulated serum was centrifuged at 3000 rpm for 10 min.

## Measurement of blood bacterial CFU

The whole blood samples were plated by 10-fold serial dilution in sterile normal saline on tryptic soy agar plates with 5% sheep blood (KisanBio, MB-B1005). Plates were incubated at 37 °C for 24 h. Then, the colonies on the plates were counted and recorded as CFU/mL.

## Western blot

The samples were electrophoresed using 10% SDS-PAGE and transferred to Immobilon-P PVDF membranes (Merck Millipore, cat. IPVH00010), which were then blocked with 5% skim milk (LPS Solution, cat. SKI500) in TBST buffer for 1 h. The membranes were

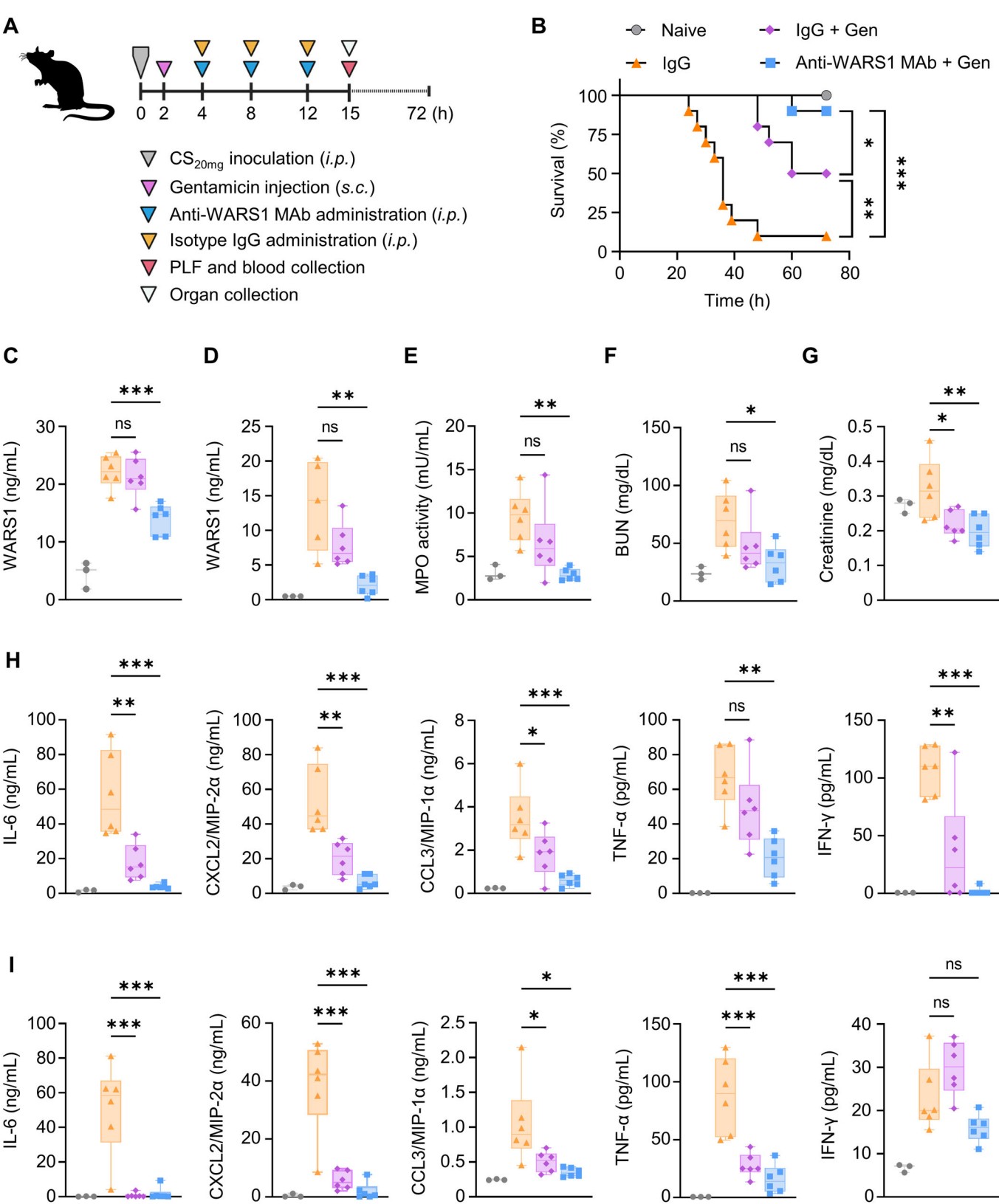

**Figure 8. WARS1 neutralization in combined with antibiotics protects severely ill septic mice from fatality.**

(A) Experimental scheme for the co-administration of antibiotics in a $CS_{20mg}$ inoculated severe septic mouse model. (B) Kaplan–Meier survival plot for mice administered with antibiotics (3 mg/kg) and control IgG or monoclonal anti-WARS1 antibody (10 mg/kg) after $CS_{20mg}$ inoculation, naïve mice were administered with PBS ($n = 10$ per group). The survival rate experiments to determine the efficacy of anti-WARS1 MAb with antibiotics were repeated two to three times. (C, D) WARS1 levels in PLF (C) and plasma (D) at 15 h after CS inoculation ($n = 5$–6 per group). (E) MPO activity in lung homogenates at 15 h after CS inoculation ($n = 6$ per group). (F, G) Serum levels of BUN (F), and creatinine (G) at 15 h after CS inoculation ($n = 6$ per group). (H, I) Levels of cytokine and chemokine in PLF (H) and plasma (I) at 15 h after CS inoculation ($n = 6$ per group). Data information: Box plots represent the median with interquartile range, the whiskers indicate min and max values (C–I). Statistical analysis is performed with log-rank test (B), and ANOVA with Bonferroni corrections (C–I). ns, not significant; *$p < 0.05$, **$p < 0.01$, ***$p < 0.001$. Source data are available online for this figure.

incubated with primary antibodies specific to WARS1 (anti-WARS1, 1:1000) at 4 °C overnight. Then, blots were incubated with goat anti-human HRP-linked secondary antibody (Emd Millipore, AP309P, 1:3000) at RT for 1 h and detected using a LAS-4000 (FujiFilm) with chemiluminescent ECL-spray (Advansta, K-12049-D50).

## Statistical analysis

Blood samples from healthy controls, ICU-controls, and patients with sepsis were analyzed using the Kruskal–Wallis test for multiple comparisons, the Mann–Whitney test for continuous variables, the log-rank test for mortality rate, the Pearson's correlation coefficient test for correlation analysis, the ROC test to calculate the WARS1 cutoff value, the multiple logistic regression test for multivariate analysis. In experiments involving mice and marmosets, for all in vitro and in vivo experiments, samples were tested in duplicates or triplicates, and survival experiments were performed two or three times, independently. The data are presented as mean ± standard deviation (SD), mean ± standard error (SEM), median with a 95% confidence interval, and min to max. Mortality analysis was performed using the log-rank test. Statistical analysis was performed using ANOVA with Bonferroni corrections for multiple comparisons, and Mann–Whitney using GraphPad Prism version 9.3.1 (GraphPad software). $p$ values <0.05 were considered statistically significant.

## Data availability

The datasets produced in RNA-seq study are available in the Gene Expression Omnibus GSE243125. Additional information required to re-analyze the reported data and information on reagents generated or used in this study are available upon request from the lead contact (mirimj@gachon.ac.kr). For more information: Mirim Jin's Website: http://iinl.gachon.ac.kr/.

## Peer review information

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

---

**The paper explained**

**Problem**
The mortality rate of sepsis remains unacceptably high; however, no specialized treatment for patients with heterogeneous sepsis currently exists owing to a lack of precise biomarkers for patient stratification and therapeutic guidance. The objective of this study was to explore the potential of WARS1 as a theranostic target for hypercytokinemic severe sepsis by examining an ICU sepsis cohort, sepsis animal models, and in vitro RNA-seq. Additionally, the study explored the application of an anti-WARS1 monoclonal antibody as a therapeutic modality.

**Results**
Highly elevated WARS1 levels can be used to stratify patients with sepsis and septic animals associated with hyperinflammation and early mortality. WARS1 substantially upregulates a cluster of genes associated with hyperinflammatory sepsis, and rWARS1 injection accelerates hypercytokinemic septic death in mice. Finally, administration of the anti-WARS1 MAb reduced fatality in severely ill septic mice by suppressing cytokine and chemokine storm.

**Impact**
A potential theranostic approach involving blood-circulating WARS1-guided patient stratification and targeted therapy using anti-WARS1 MAb may help improve the management of critically ill patients with sepsis through a precision medicine strategy.

in critically ill patients - a retrospective clinical assessment. Int J Infect Dis 97:260–266

Dobin A, Davis CA, Schlesinger F, Drenkow J, Zaleski C, Jha S, Batut P, Chaisson M, Gingeras TR (2013) STAR: ultrafast universal RNA-seq aligner. Bioinformatics 29:15–21

Hotchkiss RS, Moldawer LL, Opal SM, Reinhart K, Turnbull IR, Vincent JL (2016) Sepsis and septic shock. Nat Rev Dis Primers 2:16045

Jin M (2019) Unique roles of tryptophanyl-tRNA synthetase in immune control and its therapeutic implications. Exp Mol Med 51:1–10

Kawasaki T, Kawai T (2014) Toll-like receptor signaling pathways. Front Immunol 5:461

Kersey PJ, Allen JE, Allot A, Barba M, Boddu S, Bolt BJ, Carvalho-Silva D, Christensen M, Davis P, Grabmueller C et al (2018) Ensembl Genomes 2018: an integrated omics infrastructure for non-vertebrate species. Nucleic Acids Res 46:D802–D808

Kramer A, Green J, Pollard Jr. J, Tugendreich S (2014) Causal analysis approaches in Ingenuity Pathway Analysis. Bioinformatics 30:523–530

Kwon NH, Fox PL, Kim S (2019) Aminoacyl-tRNA synthetases as therapeutic targets. Nat Rev Drug Discov 18:629–650

Lee HC, Lee ES, Uddin MB, Kim TH, Kim JH, Chathuranga K, Chathuranga WAG, Jin M, Kim S, Kim CJ et al (2019) Released tryptophanyl-tRNA synthetase stimulates innate immune responses against viral infection. J Virol 93:e01291–18

Leligdowicz A, Matthay MA (2019) Heterogeneity in sepsis: new biological evidence with clinical applications. Crit Care 23:80

Leventogiannis K, Kyriazopoulou E, Antonakos N, Kotsaki A, Tsangaris I, Markopoulou D, Grondman I, Rovina N, Theodorou V, Antoniadou E et al (2022) Toward personalized immunotherapy in sepsis: the PROVIDE randomized clinical trial. Cell Rep Med 3:100817

Li B, Dewey CN (2011) RSEM: accurate transcript quantification from RNA-Seq data with or without a reference genome. BMC Bioinformatics 12:323

Liu SF, Malik AB (2006) NF-kappa B activation as a pathological mechanism of septic shock and inflammation. Am J Physiol Lung Cell Mol Physiol 290:L622–L645

Ma KC, Schenck EJ, Siempos II, Cloonan SM, Finkelsztein EJ, Pabon MA, Oromendia C, Ballman KV, Baron RM, Fredenburgh LE et al (2018) Circulating RIPK3 levels are associated with mortality and organ failure during critical illness. JCI Insight 3:e99692

Marshall JC (2008) Sepsis: rethinking the approach to clinical research. J Leukoc Biol 83:471–482

Marshall JC (2014) Special issue: Sepsis: why have clinical trials in sepsis failed? Trends Mol Med 20:195–203

Mei J, Liu Y, Dai N, Favara M, Greene T, Jeyaseelan S, Poncz M, Lee JS, Worthen GS (2010) CXCL5 regulates chemokine scavenging and pulmonary host defense to bacterial infection. Immunity 33:106–117

Mercer PF, Williams AE, Scotton CJ, Jose RJ, Sulikowski M, Moffatt JD, Murray LA, Chambers RC (2014) Proteinase-activated receptor-1, CCL2, and CCL7 regulate acute neutrophilic lung inflammation. Am J Respir Cell Mol Biol 50:144–157

Meyer NJ, Reilly JP, Anderson BJ, Palakshappa JA, Jones TK, Dunn TG, Shashaty MGS, Feng R, Christie JD, Opal SM (2018) Mortality benefit of recombinant human interleukin-1 receptor antagonist for sepsis varies by initial interleukin-1 receptor antagonist plasma concentration. Crit Care Med 46:21–28

Nelson M, Loveday M (2014) Exploring the innate immunological response of an alternative nonhuman primate model of infectious disease; the common marmoset. J Immunol Res 2014:913632

Nguyen TTT, Choi YH, Lee WK, Ji Y, Chun E, Kim YH, Lee JE, Jung HS, Suh JH, Kim S et al (2023) Tryptophan-dependent and -independent secretions of

tryptophanyl- tRNA synthetase mediate innate inflammatory responses. Cell Rep 42:111905

Nguyen TTT, Yoon HK, Kim YT, Choi YH, Lee WK, Jin M (2020) Tryptophanyl-tRNA synthetase 1 signals activate TREM-1 via TLR2 and TLR4. Biomolecules 10:1283

Paudel S, Baral P, Ghimire L, Bergeron S, Jin L, DeCorte JA, Le JT, Cai S, Jeyaseelan S (2019) CXCL1 regulates neutrophil homeostasis in pneumonia-derived sepsis caused by Streptococcus pneumoniae serotype 3. Blood 133:1335–1345

Peters Van Ton AM, Kox M, Abdo WF, Pickkers P (2018) Precision immunotherapy for sepsis. Front Immunol 9:1926

R Core Team (2013) R: a language and environment for statistical computing. R Foundation for Statistical Computing, Vienna

Reano A, Richard MH, Denoroy L, Viac J, Benedetto JP, Schmitt D (1993) Gamma interferon potently induces tryptophanyl-tRNA synthetase expression in human keratinocytes. J Invest Dermatol 100:775–779

Robinson MD, McCarthy DJ, Smyth GK (2010) edgeR: a Bioconductor package for differential expression analysis of digital gene expression data. Bioinformatics 26:139–140

Rubin BY, Anderson SL, Xing L, Powell RJ, Tate WP (1991) Interferon induces tryptophanyl-tRNA synthetase expression in human fibroblasts. J Biol Chem 266:24245–24248

Rudd KE, Johnson SC, Agesa KM, Shackelford KA, Tsoi D, Kievlan DR, Colombara DV, Ikuta KS, Kissoon N, Finfer S et al (2020) Global, regional, and national sepsis incidence and mortality, 1990-2017: analysis for the Global Burden of Disease Study. Lancet 395:200–211

Scicluna BP, van Vught LA, Zwinderman AH, Wiewel MA, Davenport EE, Burnham KL, Nurnberg P, Schultz MJ, Horn J, Cremer OL et al (2017) Classification of patients with sepsis according to blood genomic endotype: a prospective cohort study. Lancet Respir Med 5:816–826

Seymour CW, Kennedy JN, Wang S, Chang CH, Elliott CF, Xu Z, Berry S, Clermont G, Cooper G, Gomez H et al (2019) Derivation, validation, and potential treatment implications of novel clinical phenotypes for sepsis. JAMA 321:2003–2017

Shalova IN, Lim JY, Chittezhath M, Zinkernagel AS, Beasley F, Hernandez-Jimenez E, Toledano V, Cubillos-Zapata C, Rapisarda A, Chen J et al (2015) Human monocytes undergo functional re-programming during sepsis mediated by hypoxia-inducible factor-1alpha. Immunity 42:484–498

Singer M, Deutschman CS, Seymour CW, Shankar-Hari M, Annane D, Bauer M, Bellomo R, Bernard GR, Chiche JD, Coopersmith CM et al (2016) The Third International Consensus Definitions for Sepsis and Septic Shock (Sepsis-3). JAMA 315:801–810

Soares MP, Teixeira L, Moita LF (2017) Disease tolerance and immunity in host protection against infection. Nat Rev Immunol 17:83–96

Souto FO, Alves-Filho JC, Turato WM, Auxiliadora-Martins M, Basile-Filho A, Cunha FQ (2011) Essential role of CCR2 in neutrophil tissue infiltration and multiple organ dysfunction in sepsis. Am J Respir Crit Care Med 183:234–242

Stanski NL, Wong HR (2020) Prognostic and predictive enrichment in sepsis. Nat Rev Nephrol 16:20–31

Starr ME, Steele AM, Saito M, Hacker BJ, Evers BM, Saito H (2014) A new cecal slurry preparation protocol with improved long-term reproducibility for animal models of sepsis. PLoS ONE 9:e115705

Subramanian A, Tamayo P, Mootha VK, Mukherjee S, Ebert BL, Gillette MA, Paulovich A, Pomeroy SL, Golub TR, Lander ES et al (2005) Gene set enrichment analysis: a knowledge-based approach for interpreting genome-wide expression profiles. Proc Natl Acad Sci USA 102:15545–15550

Sweeney TE, Azad TD, Donato M, Haynes WA, Perumal TM, Henao R, Bermejo-Martin JF, Almansa R, Tamayo E, Howrylak JA et al (2018) Unsupervised

analysis of transcriptomics in bacterial sepsis across multiple datasets reveals three robust clusters. Crit Care Med 46:915–925

't Hart BA, Laman JD, Bauer J, Blezer E, van Kooyk Y, Hintzen RQ (2004) Modelling of multiple sclerosis: lessons learned in a non-human primate. Lancet Neurol 3:588–597

## Acknowledgements

This research was supported by the Bio & Medical Technology Development Program of the National Research Foundation (NRF) of the Korean government (MSIT), Republic of Korea, grant numbers NRF-2019M3E5D5064771, the Korea Health Technology R&D Project by the Korea Health Industry Development Institute (KHIDI) of the Ministry of Health & Welfare, Republic of Korea, grant number HI20C0015, and HI22C1883, Daegu-Gyeongbuk/Osong Medical Cluster R&D Project funded by the Ministry of Science and ICT, the Ministry of Trade, Industry and Energy, the Ministry of Health & Welfare, Republic of Korea, grant number HI19C0763, and Korea Drug Development Fund funded by Ministry of Science and ICT, Ministry of Trade, Industry, and Energy, and Ministry of Health and Welfare, Republic of Korea, grant number RS-2022-00166575.

## Author contributions

**Yoon Tae Kim**: Conceptualization; Data curation; Formal analysis; Validation; Investigation; Visualization; Methodology; Writing—original draft; Writing—review and editing. **Jin Won Huh**: Conceptualization; Resources; Formal analysis; Investigation; Methodology; Writing—review and editing. **Yun Hui Choi**: Resources; Formal analysis; Methodology. **Hee Kyeong Yoon**: Investigation. **Tram TT Nguyen**: Conceptualization; Investigation; Writing—review and editing. **Eunho Chun**: Investigation. **Geunyeol Jeong**: Investigation. **Sunyoung Park**: Conceptualization; Investigation. **Sungwoo Ahn**: Formal analysis. **Won Kyu Lee**: Resources; Funding acquisition; Investigation. **Young-Woock Noh**: Formal analysis; Investigation. **Kyoung Sun Lee**: Formal analysis; Investigation. **Hee Sung Ahn**: Investigation. **Cheolju Lee**: Formal analysis. **Sang Min Lee**: Resources; Methodology. **Kyung Su Kim**: Resources; Formal analysis; Investigation; Methodology; Writing—review and editing. **Gil Joon Suh**: Formal analysis. **Kyeongman Jeon**: Formal analysis. **Sunghoon Kim**: Writing—review and editing. **Mirim Jin**: Conceptualization; Supervision; Funding acquisition; Writing—original draft; Writing—review and editing.

## Disclosure and competing interests statement

MJ is the founder and a shareholder of MirimGENE Co., Ltd., Incheon, South Korea. The other authors declare no competing interests.

# Expanded View Figures

**A**

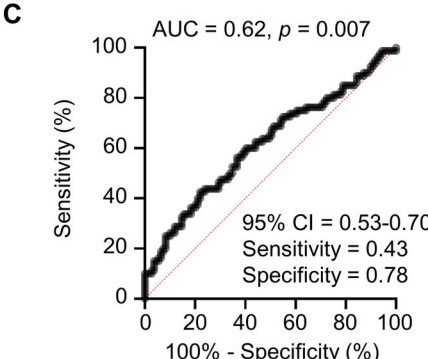

**B**

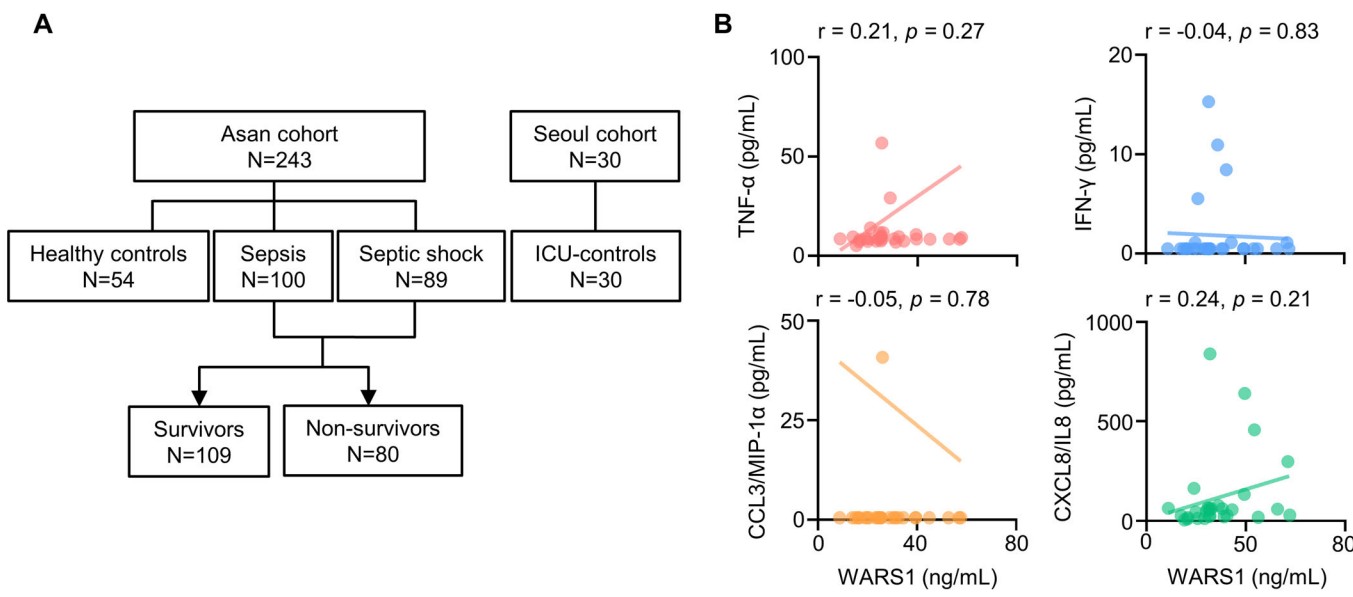

**C**

**Figure EV1. Sepsis and ICU cohort study.**

(**A**) Flowchart of the sepsis and ICU cohort population in Asan Medical Center and Seoul National University Hospital. (**B**) Correlation between WARS1 levels and cytokine and chemokine levels in the ICU controls ($n = 30$). Individual correlation results are reported with linear regression lines. (**C**) Receiver operating characteristic (ROC) analysis by cut-off value of WARS1 between survivors and non-survivors. Data information: Statistical analysis is performed with Pearson's correlation coefficient test (**B**), and ROC test (**C**).

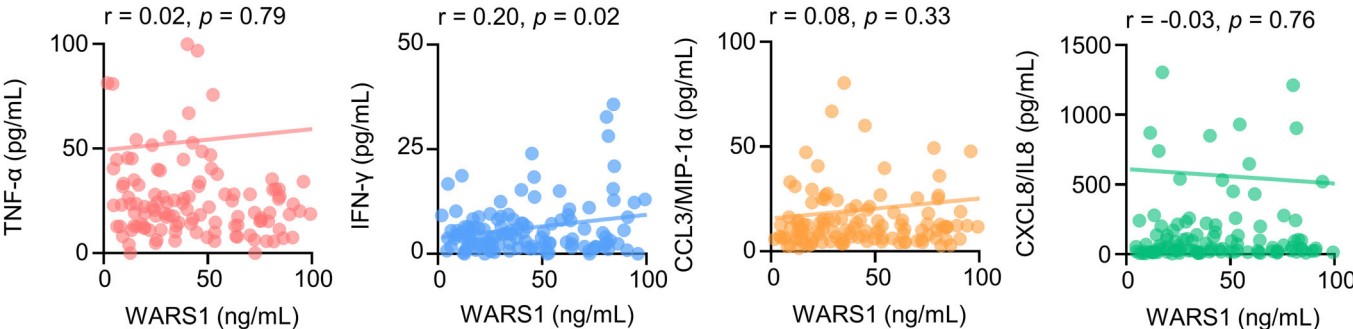

**Figure EV2. The WARS1<sup>low</sup> group showed no positive correlation with cytokine and chemokine levels.**

Correlation between WARS1 levels and cytokine and chemokine levels in the WARS1<sup>low</sup> ($n = 130$) group (stratified below 106.3 ng/mL). Individual correlation results are reported with linear regression lines. Statistical analysis is performed with Pearson's correlation coefficient test.

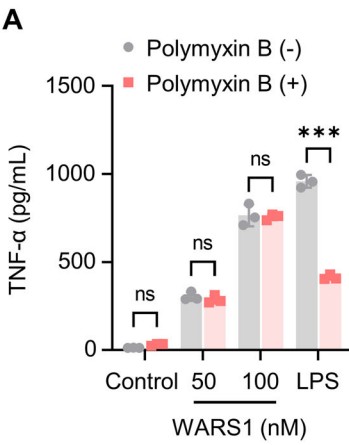

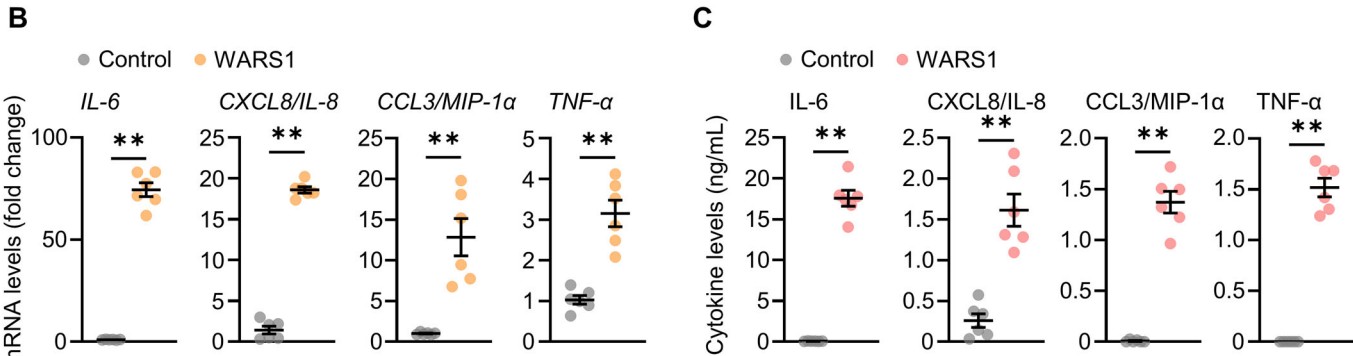

**Figure EV3. Effect of WARS1-induced pro-inflammatory response in hPBMCs.**

(A) TNF-α levels in the supernatant of PMA-differentiated THP-1 cells ($n = 3$) treated with hFL-WARS1 (50–100 nM) or LPS (100 ng/mL) with or without polymyxin B (50 μg/mL) for 9 h. (B) Gene expression of cytokine and chemokine in hPBMCs ($n = 6$) treated with hFL-WARS1 (50 nM) for 6 h. (C) Levels of cytokine and chemokine in the supernatants of hPBMCs ($n = 6$) treated with hFL-WARS1 (50 nM) for 6 h. Data information: Data are presented as mean ± SD (A), and ± SEM (B, C). Statistical analysis is performed with ANOVA with Bonferroni corrections (A), and Mann–Whitney $U$-test (B, C). ns, not significant; **$p < 0.01$, ***$p < 0.001$.

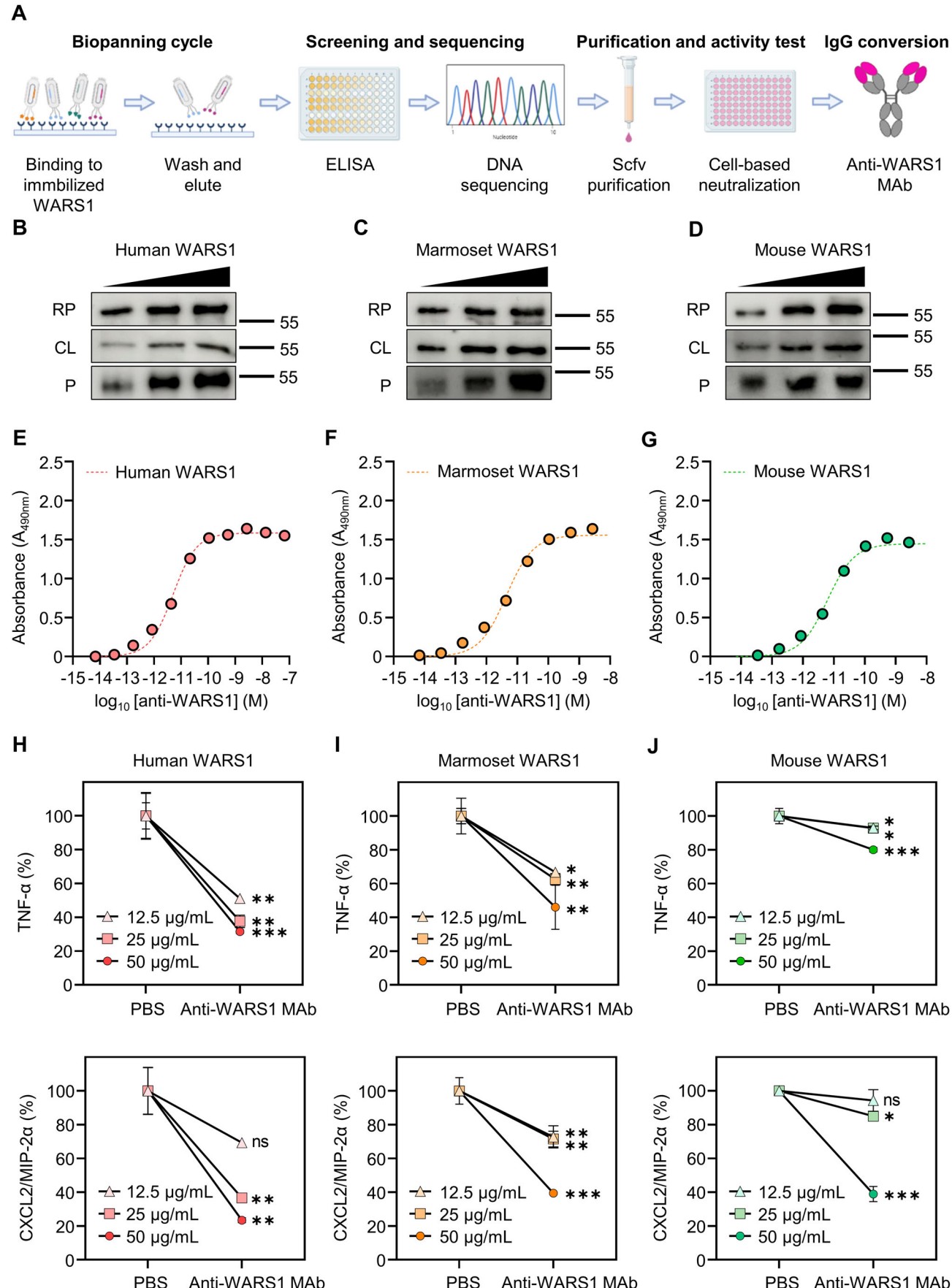

**Figure EV4.   Generation and neutralizing effect of anti-WARS1 MAb.**

(A) Production process of WARS1 monoclonal antibody. After selectively isolating a clone that binds to WARS1 from the phage library (biopanning cycle), affinity and specificity were assessed, followed by DNA screening and sequencing. Then, the purified scfv was used for the neutralization test (purification and activity test) and finally converted to IgG (IgG conversion). (B–D) Immunoblot for human (B), marmoset (C), and mouse (D) WARS1 with anti-WARS1 MAb in recombinant full-length protein (RP), cell lysate (CL), and plasma (P). (E–G) Binding affinity of anti-WARS1 MAb to recombinant human (E), marmoset (F), and mouse (G) WARS1. (H–J) Levels of TNF-α and CXCL2/MIP-2α in the supernatant. J774A.1 cells ($n = 3$) were treated with a mixture of human (H), marmoset (I), and mouse (J) WARS1 and anti-WARS1 MAb or isotype IgG for 16 h.  Data information: Data are presented as mean ± SD (H–J). Statistical analysis is performed with ANOVA with Bonferroni corrections (H–J). ns, not significant; *$p < 0.05$, **$p < 0.01$, ***$p < 0.001$.

**A**

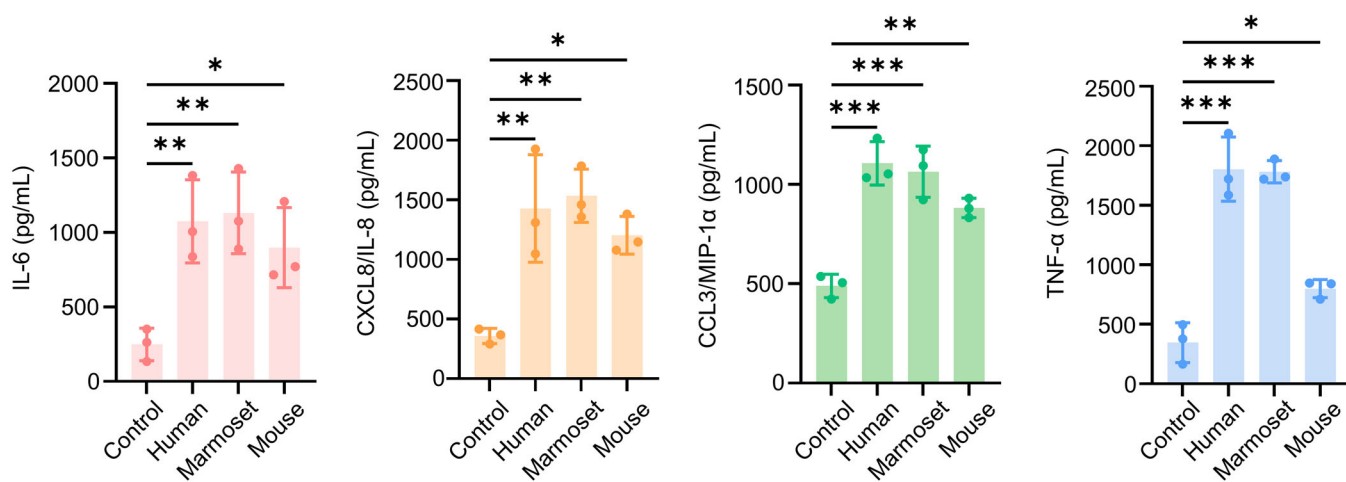

Human
Marmoset
Mouse

---MPNSEXCASPLELFNSIATQGELVRSLKAGNASKDEIDSAVKMLLSLKMSYKAAMGEDYKAXCPPGNPAPXSNHGPDA--TEAEEDFVDPWTVQTSS

---MPNSEP-ASLLELFNSIATQGELVRSLKAGNASKDEIDSAVKMLVSLKMSYKAAAGEDYKADCPPGNPAPTSNHGPDA--TEAEEDFVDPWTVQTSS 94
---MANSEACASPLELFNSVTTQGEHVRALKAGNASKDEIDSAVKMLLSLKMSYKAAMGEDYKANCPPGSLAPSSNHGPDAMITEAEEDFVDPWTVQTSS 97
MADMPSGESCTSPLELFNSIATQGELVRSLKAGNAPKDEIDSAVKMLLSLKMSYKAAMGEEYKAGCPPGNPTAGRNCDSDA--TKASEDFVDPWTVRTSS 98

Human
Marmoset
Mouse

AKGIDYDKLIVRFGSSKIDKELINRIERATGQRPHHFLRRGIFFSHRDMNQVLDAYENKKPFYLYTGRGPSSEAMHVGHLIPFIFTKWLQDVFNVPLVIQ

AKGIDYDKLIVRFGSSKIDKELINRIERATGQRPHHFLRRGIFFSHRDMNQVLDAYENKKPFYLYTGRGPSSEAMHVGHLIPFIFTKWLQDVFNVPLVIQ 194
AKGIDYDKLIVRFGSSKIDKELINRIERATGQRPHHFLRRGIFFSHRDMNQVLDAYENKKPFYLYTGRGPSSEAMHVGHLIPFIFTKWLQDVFNVPLVIQ 197
AKGIDYDKLIVQFGSSKIDKELINRIERATGQRPHRFLRRGIFFSHRDMNQILDAYENKKPFYLYTGRGPSSEAMHLGHLVPFIFTKWLQDVFNVPLVIQ 198

Human
Marmoset
Mouse

MTDDEKYLWKDLTLDQAYSYAVENAKDIIACGFDINKTFIFSDLDYMGMSPGFYKNVVKIQKHVTFNQVKGIFGFTDSDCIGKISFPAIQAAPSFSNSFP

MTDDEKYLWKDLTLDQAYSYAVENAKDIIACGFDINKTFIFSDLDYMGMSSGFYKNVVKIQKHVTFNQVKGIFGFTDSDCIGKISFPAIQAAPSFSNSFP 294
MTDDEKYLWKDLTLDQAYGYAVENAKDIIACGFDINKTFIFSDLDYMGMSPGFYKNVVKIQKHVTFNQVKGIFGFTDSDSIGKISFPAIQAAPSFSNSFP 297
MSDDEKYLWKDLTLEQAYSYTVENAKDIIACGFDINKTFIFSDLEYMGQSPGFYRNVVKIQKHVTFNQVKGIFGFTDSDCIGKISFPAVQAAPSFSNSFP 298

Human
Marmoset
Mouse

QIFRDRTDIQCLIPCAIDQDPYFRMTRDVAPRIGYPKPALLHSTFFPALQGAQTKMSASDPNSSIFLTDTAKQIKTKVNKHAFSGGRDTVEEHRQFGGNC

QIFRDRTDIQCLIPCAIDQDPYFRMTRDVAPRIGYPKPALLHSTFFPALQGAQTKMSASDPNSSIFLTDTAKQIKTKVNKHAFSGGRDTIEEHRQFGGNC 394
QIFGDRTDIQCLIPCAIDQDPYFRMTRDVAPKIGYPKPALLHSTFFPALQGAQTKMSASDPNSSIFLTDTAKQIKTKVNKHAFSGGRDTVEEHRQFGGNC 397
KIFRDRTDIQCLIPCAIDQDPYFRMTRDVAPRIGHPKPALLHSTFFPALQGAQTKMSASDPNSSIFLTDTAKQIKSKVNKHAFSGGRDTVEEHRQFGGNC 398

Human
Marmoset
Mouse

DVDVSFMYLTFFLEDDDKLEQIRKDYTSGAMLTGELKKTLIEVLQPLIAEHQARRKEVTDEIVKEFMTPRKLSFDFQ------

DVDVSFMYLTFFLEDDDKLEQIRKDYTSGAMLTGELKKALIEVLQPLIAEHQARRKEVTDEIVKEFMTPRKLSFDFQ 471
DVDVSFMYLTFFLEDDDKLEQIRKDYTSGAMLTGELKKTLIEVLQPLIAEHQARRKEVTDEIVKEFMTPRKLSFDFQ 474
EVDVSFMYLTFFLEDDDRLEQIRKDYTSGAMLTGELKKTLIDVLQPLIAEHQARRKAVTEETVKEFMTPRQLSFHFQCFCFDT 481

**B**

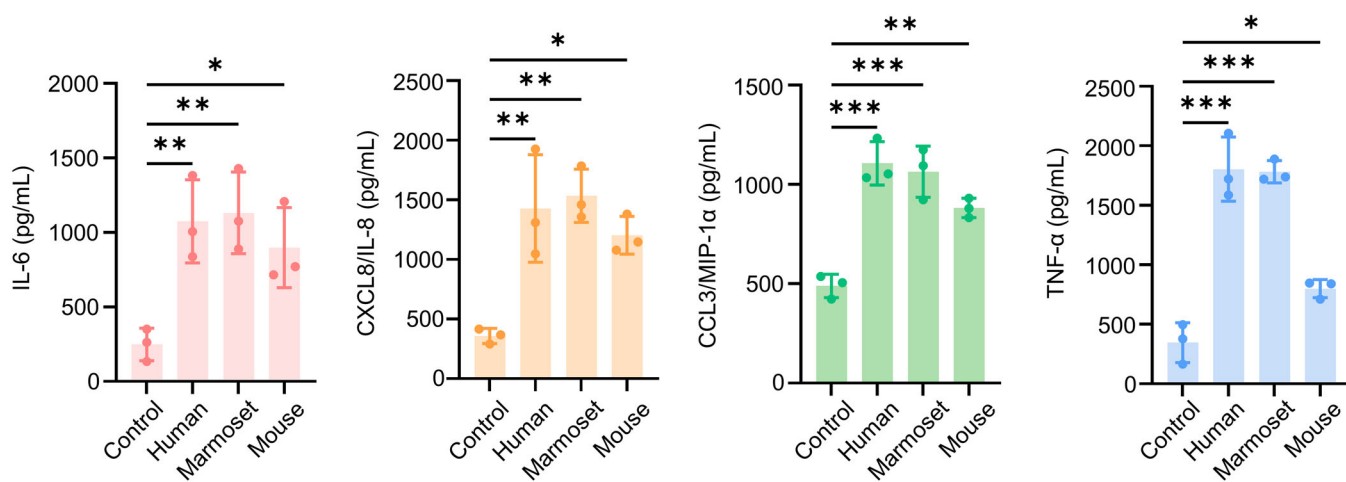

**Figure EV5. Homology between human, marmoset, and mouse WARS1.**

(A) Similarity of protein sequence between human, marmoset, and mouse FL-WARS1. (B) Levels of cytokine and chemokine in the supernatant of PMA-differentiated THP-1 cells ($n = 3$) treated with human, marmoset, and mouse WARS1 (50 nM) for 18 h. Data information: Data are presented as mean ± SD (B). Statistical analysis is performed with ANOVA with Bonferroni corrections (B). *$p < 0.05$, **$p < 0.01$, ***$p < 0.001$.

