## [Peer Review File · EMBO Molecular Medicine]

Highly secreted tryptophanyl tRNA synthetase 1 as a potential theranostic target for hypercytokinemic severe sepsis

Yoon Tae Kim, Jin Won Huh, Yun Hui Choi, Hee Kyeong Yoon, Tram Nguyen, Eunho Chun, Geunyeol Jeong, Sunyoung Park, Sungwoo Ahn, Won-Kyu Lee, Young-Woock Noh, Kyoung Sun Lee, Hee Sung Ahn, Cheolju Lee, Sang Min Lee, Kyung Su Kim, Gil Joon Suh, Kyeongman Jeon, Sunghoon Kim, and Mirim Jin

DOI: [10.15252/emmm.202318106](https://doi.org/10.15252/emmm.202318106)

Corresponding author: *Mirim Jin (mirimj@gachon.ac.kr)*

Review Timeline:

Submission Date:	1st Jun 23
Editorial Decision:	7th Jul 23
Revision Received:	17th Oct 23
Editorial Decision:	26th Oct 23
Revision Received:	2nd Nov 23
Editorial Decision:	6th Nov 23
Revision Received:	9th Nov 23
Accepted:	10th Nov 23

Editor: *Zeljko Durdevic*

Transaction Report:

7th Jul 2023

Dear Prof. Jin,

Thank you for the submission of your manuscript to EMBO Molecular Medicine. We have now received feedback from the three reviewers who agreed to evaluate your manuscript. While the referee #1 is overall supportive and raises important but minor criticism, referees #2 and #3 recognize potential interest of the study but also raise serious concerns that should be addressed in a major revision. Our cross-commenting session made it clear that main conclusions are not sufficiently supported by the data. Therefore, inclusion of non-septic ICU control patients as suggested by the referee #2 and addressing referee's #3 main points experimentally is required for further consideration of your manuscript. Addressing referee's #3 points in "Further comments and questions" experimentally is of course welcomed but not essential.

Considering that the revision will require extensive experimentations we think six months rather than three months would be more appropriate to provide the complete revision. If you would like to discuss further the points raised by the referees, I am available to do so via email or video. Let me know if you are interested in this option.

We would welcome the submission of a revised version within six months for further consideration. Please let us know if you require longer to complete the revision.

I look forward to receiving your revised manuscript.

Yours sincerely,

Zeljko Durdevic

We require:

- 1) A .docx formatted version of the manuscript text (including legends for main figures, EV figures and tables). Please make sure that the changes are highlighted to be clearly visible.
- 2) Individual production quality figure files as .eps, .tif, .jpg (one file per figure). For guidance, download the 'Figure Guide PDF': (<https://www.embopress.org/page/journal/17574684/authorguide#figureformat>).
- 3) A .docx formatted letter INCLUDING the reviewers' reports and your detailed point-by-point responses to their comments. As part of the EMBO Press transparent editorial process, the point-by-point response is part of the Review Process File (RPF), which will be published alongside your paper.
- 4) A complete author checklist, which you can download from our author guidelines (<https://www.embopress.org/page/journal/17574684/authorguide#submissionofrevisions>). Please insert information in the

checklist that is also reflected in the manuscript. The completed author checklist will also be part of the RPF.

6) It is mandatory to include a 'Data Availability' section after the Materials and Methods. Before submitting your revision, primary datasets produced in this study need to be deposited in an appropriate public database, and the accession numbers and database listed under 'Data Availability'. Please remember to provide a reviewer password if the datasets are not yet public (see <https://www.embopress.org/page/journal/17574684/authorguide#dataavailability>).

.

13) Author contributions: You will be asked to provide CRediT (Contributor Role Taxonomy) terms in the submission system. These replace a narrative author contribution section in the manuscript.

14) A Conflict of Interest statement should be provided in the main text.

Please note: When submitting your revision you will be prompted to enter your funding and payment information. This will allow Wiley to send you a quote for the article processing charge (APC) in case of acceptance. This quote takes into account any reduction or fee waivers that you may be eligible for. Authors do not need to pay any fees before their manuscript is accepted and transferred to the publisher.

EMBO Press participates in many Publish and Read agreements that allow authors to publish Open Access with reduced/no publication charges. Check your eligibility: <https://authorservices.wiley.com/author-resources/Journal-Authors/open-access/affiliation-policies-payments/index.html>

***** Reviewer's comments *****

Referee #1 (Comments on Novelty/Model System for Author):

All experiments are appropriate and well designed
The authors provide a tremendous amount of work ending to a rather convincing conclusion.

Referee #1 (Remarks for Author):

Yoon Tae Kim et colleagues further extend their previous observation on the involvement of tryptophanyl-tRNA synthetase in host defense and inflammation. They now propose an amazing amount of works to demonstrate that this enzyme is a biomarker of severity in human sepsis and in a murine model of sepsis, acts as a key player in cell activation, cytokine production, organ dysfunction and death. They also showed its deleterious synergistic effect with infection. The experiments with the Mab are excellent and the authors have to be congratulated for their work.

Their data are quite convincing, even if some are puzzling.

- The action of tryptophanyl-tRNA synthetase through TLR4/MD2. Even if the controls have been done and are fine, it is known that polymyxin B does not inhibit all endotoxins (DOI: 10.1016/0161-5890(86)90127-6) and the sensitivity of the LAL test also depends on the origin of the LPS (DOI: 10.1016/1043-4666(90)90025-o). This remark is not specific to tryptophanyl-tRNA synthetase, but it unclear how the TLR4/MD2 receptor can bind so many biochemically diverse ligands.
- But my main question concerns whether the observation is specific for tryptophanyl-tRNA synthetase or whether similar results (or not) could be obtained with any of the 19 other aminoacyl-tRNA synthetases. Why would it be specific to this enzyme? It would have been great to test at least one or two of them. Please discuss this aspect.

Minor comments

1. When a r value is below 0.5, it means that there is no true correlation, so some of the figures on correlations are poorly convincing (fig.1 & 2, EV2).
2. It is unclear why GSEA analysis reveals the activation of the TLR signaling while IPA does not.
3. It is unclear why the anti-WARS1 Mab Prevent IL-6 mRNA expression but not the protein expression (fi.5C & F).
4. Please provide AUC for fig. EV1

Referee #2 (Comments on Novelty/Model System for Author):

"healthy controls" reflect a questionable control / patients hospitalized with or without organ dysfunction in the absence of infection are warranted!

Referee #2 (Remarks for Author):

Despite decades of research there are no treatment options for individualized immunotherapy in human sepsis.

The authors aim to identify a novel strategy to combat overwhelming immune response during sepsis to improve outcome, by addressing an integrative approach combining data obtained either from a very well characterized cohort of patients with sepsis and septic shock as well as from two sophisticated animal models (rodents/marmots) with escalating severity/antibiotic rescue, all in all fulfilling Koch's postulates to establish a causal relationship between pathogens, immune response of the host and outcome.

Results from the study are of great interest for the field of molecular sepsis research, since the potential to control a novel mediator released from activated immune cells (termed WARS1) was identified, which seems involved in boosting immune responses. In patients with sepsis and septic shock, a severity dependent increase was identified with a significant association with respect to prediction of unfavorable outcome by definition of a cut-off level. In the animal models, a similar course of plasma WARS1 was observed, and neutralization and combination with antibiotic rescue protects animals with severe sepsis from mortality by dampening excessive cytokine/chemokine release and organ damage. All in all the paper is well written and of great interest of the scientific community. Animal models, time points of sampling and methods are well selected and described. However, there are some major and minor concerns as given below to improve quality:

Major concerns:

- The cohort of patients with sepsis/septic shock is only compared to healthy controls. It is of great interest, whether this mechanism is specific to infection sepsis/septic shock (i.e. organ dysfunction and shock - e.g. due to trauma, surgery) are behaving differently. A complementary analysis of appropriate ICU-controls (in addition to healthy controls) is required to improve the quality of your data set.
- Following mAb administration the bacterial count is unchanged (Fig 6P). This potential mechanism of "disease tolerance" should be addressed in detail in the discussion section.
- Microphotographs of organ destruction are too small and do not allow for conclusive interpretation (Fig. 6)
- Did you perform the animal experiments in a blinded fashion? Description is needed or also a justification for waiving this. Same is true for histological examination.

Minor concerns:

- There are some semi-cola throughout the Manuscript, which do not improve the understanding (Abstract line 7 (71) and others).
- N-terminal active domain, use the italic notation along current guidelines.

Referee #3 (Comments on Novelty/Model System for Author):

1. The used methods are appropriate.
2. It is well known that the cytokine storm is a relevant feature of sepsis and septic shock associated with high mortality. The role of WARS1 in this respect is not fully understood. Several upstream mediators have been described that might be a useful target in treating sepsis.
3. So far, the results of applying anti-WARS1 indicate a beneficial effect during sepsis, however, how to identify the patients at risk and possibly benefiting from anti-WARS1 treatment? It is still unknown what possible effects the treatment with anti-WARS1 might have in respect to secondary viral infections or viral reactivation. This still needs to be answered in further studies.
4. The model system is adequate.

Referee #3 (Remarks for Author):

The authors investigated the role of WARS1 in sepsis using a translational approach and designed an anti-WARS1-neutralizing antibody showing beneficial effects during sepsis. The manuscript is nicely written.

Main questions:

1. How important is WARS1 in sepsis compared to septic shock? Although the low WARS1 group (Fig. EV2) shows no significant correlation with cytokines and chemokines, a still significant fraction dies from sepsis (Fig. 1C). Can the authors explain this experimentally and discuss in more detail when anti-WARS1 might be beneficial during sepsis? How robust would the discrimination into WARS1 low vs. high group be that identifies patients possibly benefiting from anti-WARS1 treatment?

2. Does anti-WARS1 treatment increase the risk for secondary viral infections during sepsis?

Further comments and questions:

1. WARS1 is secreted by monocytes. It has been shown that IL-3 is an upstream regulator of inflammation during sepsis and that plasma IL-3 levels are positively correlated with circulating monocytes. Please provide data on the IL-3 levels in your sepsis cohorts. Is there a correlation of IL-3 and WARS1? Can the authors discuss a possible link between IL-3, monocytes and WARS1, if a correlation can be detected.
2. Can the authors subgroup the various monocyte population using CD14 and CD16 and perform correlation with WARS1?
3. Do WARS1 levels correlate with monocyte apoptosis?
4. I do not understand yet why the authors define WARS1 as upstream regulator. Please provide mechanistic data proving that WARS1 indeed induces the highly complex inflammatory signaling pathways leading to monocytosis, neutrophilia, and elevated cytokine and chemokine levels. So far the authors only provide correlative data. Of note, WARS1 is produced by monocytes, and higher WARS1 levels are correlated with higher monocytes and neutrophils, which are main producers of many of the mentioned cyto- and chemokines. But who induces high monocytes and neutrophils? IL-3?
5. Please provide apoptosis data on human monocytic THP-1 cells after rWARS1 treatment.
6. What happens to the cytokine, chemokine and leukocyte levels when rWARS1 is injected i.v. into naive mice?
7. What happens if anti-WARS1 is administered prior to LPS or CS injection?

Point by-point response to “EMM-2023-18106”

We really appreciate the reviewers' efforts to improve our manuscript. We have attempted to address all reviewers' valuable comments and suggestions and hope our responses meet your requirements. The revised manuscript has been annotated with red highlights.

Referee #1 (Comments on Novelty/Model System for Author):

All experiments are appropriate and well designed

The authors provide a tremendous amount of work ending to a rather convincing conclusion.

Referee #1 (Remarks for Author):

Yoon Tae Kim et colleagues further extend their previous observation on the involvement of tryptophanyl-tRNA synthetase in host defense and inflammation. They now propose an amazing amount of works to demonstrate that this enzyme is a biomarker of severity in human sepsis and in a murine model of sepsis, acts as a key player in cell activation, cytokine production, organ dysfunction and death. They also showed its deleterious synergistic effect with infection. The experiments with the Mab are excellent and the authors have to be congratulated for their work. Their data are quite convincing, even if some are puzzling.

Major comments

The action of tryptophanyl-tRNA synthetase through TLR4/MD2. Even if the controls have been done and are fine, it is known that polymyxin B does not inhibit all endotoxins (DOI: 10.1016/0161-5890(86)90127-6) and the sensitivity of the LAL test also depends on the origin of the LPS (DOI: 10.1016/1043-4666(90)90025-o).

Answer: We appreciate this reviewer's valuable comment and understand the concerns raised regarding endotoxin contamination. In our studies, we confirmed negligible endotoxin levels in purified proteins by using an endotoxin detection kit, and polymyxin B treatment. We also used the endotoxin-free recombinant proteins isolated from HEK293T cell culture and observed that FL-WARS1, but not mini-WARS1, obtained from the mammalian cells also showed the capability of activating TLR4/MD2 (Ahn *et al*, 2017).

This remark is not specific to tryptophanyl-tRNA synthetase, but it is unclear how the

TLR4/MD2 receptor can bind so many biochemically diverse ligands.

Answer: We appreciate this reviewer's radical and intriguing question. We believe that a substantial amount of independent research would be required to provide an answer. At least, we would like to remind that WARS1 would bind to TLR4/MD2 in a manner distinguished from LPS (Ahn *et al.*, 2017).

But my main question concerns whether the observation is specific for tryptophanyl-tRNA synthetase or whether similar results (or not) could be obtained with any of the 19 other aminoacyl-tRNA synthetases. Why would it be specific to this enzyme? It would have been great to test at least one or two of them. Please discuss this aspect.

Answer: We appreciate this reviewer's valuable comment. In our opinion, there are a few reasons why WARS1 can work as a specific ligand for TLR4/MD2. First, WARS1 has a unique N-terminal extension containing the WHEP domain that is involved in the interaction with TLR4/MD2. Second, among human ARSs, WARS1 appears to be a specific one that is rapidly secreted in response to bacterial infection. Third, the expression of WARS1 is specifically induced by IFN-gamma, suggesting its functional significance in immune responses.

According to this reviewer's suggestion for other ARS, we measured leucyl-tRNA synthetase 1 (LARS1) levels in currently available samples from healthy controls ($n = 30$), ICU-controls ($n = 30$), and patients with sepsis ($n = 46$) and septic shock ($n = 43$). LARS1 levels were higher in patients with sepsis than in healthy controls; however, no differences were observed between ICU controls and patients with sepsis or between patients with sepsis and those with septic shock. Moreover, LARS1 levels did not differ between survivors and non-survivors. These findings further supported that the blood levels of WARS1, but not LARS1, specifically changed depending on the severity of patients with sepsis.

Due to a lack of plasma samples from some patients, not all patients with sepsis from our original manuscript could be analyzed. For this reason, we would like to provide the result for this reviewer's reference.

Figure legend

Left. Plasma levels of LARS1 (log10 scale) in healthy controls ($n = 30$), ICU-controls ($n = 30$), patients with sepsis ($n = 46$), and patients with septic shock ($n = 43$).

Right. Plasma levels of LARS1 (log10 scale) between survivors ($n = 49$) and non-survivors ($n = 40$).

Data information: Data are presented as median with 95% confidence interval. Statistical analysis is performed with Kruskal-Wallis test (left), Mann-Whitney U-test (right). ns, not significant; *** $p < 0.001$.

Minor comments

When a r value is below 0.5, it means that there is no correlation, so some of the figures on correlations are poorly convincing (fig.1 & 2, EV2).

Answer: We appreciate this reviewer pointing this out. We used the correlation analysis method following a previous study (Mukaka, 2012). In this method, 0.30~0.50 and 0.50~0.70 are described as low and moderate positive correlations, respectively. The description was revised in lines 147–150.

Furthermore, WARS1 levels in the high-WARS1 group showed a moderate positive correlation with TNF- α levels and a low positive correlation with IFN- γ , CXCL8/IL-8, and CCL3/MIP-1 α , indicating an association with proinflammatory responses (Fig 1D)

It is unclear why GSEA analysis reveals the activation of the TLR signaling while IPA does not.

Answer: We thank this reviewer's valuable comments. The FDR q -value of TLR signaling in GSEA was 0.007 ($-\log(\text{FDR } q\text{-value}) = 2.17$), indicating an activated pathway. Similarly, in

IPA, the p -value for TLR signaling was 0.04 ($-\log(p\text{-value}) = 1.35$), indicating an activated pathway. The ten pathways presented in the figure were statistically significant. We have modified the sentence in the revised manuscript (lines 200–202).

IPA also revealed the most significant changes in pattern recognition receptors (PRRs), triggering receptors expressed on myeloid cells 1 (TREM-1), IL-8 signaling, and TLR signaling ($-\log(p\text{-value}) > 1$, Fig 3G).

It is unclear why the anti-WARS1 Mab Prevent IL-6 mRNA expression but not the protein expression (fi.5C & F).

Answer: A previous study (Lu *et al*, 2012) reported that IL-6 is promptly released from LPS-stimulated endothelial cells. Therefore, we think that it cannot be removed by anti-WARS1 MAb, although anti-WARS1 MAb can suppress WARS1-induced IL-6 mRNA expression and release from monocytes and macrophages in the blood.

Please provide AUC for fig. EV1.

Answer: We have presented the AUC in Fig EV1E.

Referee #2 (Comments on Novelty/Model System for Author):

"healthy controls" reflect a questionable control / patients hospitalized with or without organ dysfunction in the absence of infection are warranted!

Answer: We appreciate the reviewers' valuable request. We further examined the ICU control cohort that was comprised of 30 patients who were resuscitated from cardiac arrest with no infection. Please see our responses in the Major concerns section.

Referee #2 (Remarks for Author):

Despite decades of research there are no treatment options for individualized immunotherapy in human sepsis.

The authors aim to identify a novel strategy to combat overwhelming immune response during sepsis to improve outcome, by addressing an integrative approach combining data obtained either from a very well characterized cohort of patients with sepsis and septic shock

as well as from two sophisticated animal models (rodents/marmots) with escalating severity/antibiotic rescue, all in all fulfilling Koch's postulates to establish a causal relationship between pathogens, immune response of the host and outcome.

Results from the study are of great interest for the field of molecular sepsis research, since the potential to control a novel mediator released from activated immune cells (termed WARS1) was identified, which seems involved in boosting immune responses. In patients with sepsis and septic shock, a severity dependent increase was identified with a significant association with respect to prediction of unfavorable outcome by definition of a cut-off level. In the animal models, a similar course of plasma WARS1 was observed, and neutralization and combination with antibiotic rescue protects animals with severe sepsis from mortality by dampening excessive cytokine/chemokine release and organ damage. All in all the paper is well written and of great interest of the scientific community. Animal models, time points of sampling and methods are well selected and described. However, there are some major and minor concerns as given below to improve quality:

Major concerns

The cohort of patients with sepsis/septic shock is only compared to healthy controls. It is of great interest, whether this mechanism is specific to infection sepsis/septic shock (i.e. organ dysfunction and shock - e.g. due to trauma, surgery) are behaving differently. A complementary analysis of appropriate ICU-controls (in addition to healthy controls) is required to improve the quality of your data set.

Answer: We appreciate this reviewer's critical request for patients' sample study. We prospectively collected plasma samples from a cohort of adult patients admitted to the intensive care unit after successful resuscitation from non-traumatic cardiac arrest at Seoul National University Hospital. We have described the results in lines 135–137 and in the Materials and Methods section (lines 366–374) as follows:

Furthermore, we included ICU controls ($n = 30$), which comprised adult patients admitted to the ICU of Seoul National University Hospital following successful resuscitation from non-traumatic cardiac arrest with no infection (Fig EV1A and B).

The prospective biobank of post-cardiac arrest patients was analyzed as part of the ICU

control group. The Institutional Review Board approved the prospective patient enrollment and sample preservation (SNUH IRB No. 1707-012-865) as well as this analysis (SNUH IRB No. 2307-149-1452). The study protocol was registered at ClinicalTrials.gov (NCT01670383). Adult patients admitted to the ICU of the study hospital following successful resuscitation from nontraumatic cardiac arrest were prospectively screened. Blood samples were collected at admission and 24 h after admission after obtaining written informed consent from the patient or next of kin. Serum was separated and stored at -80 °C until analysis. A total of 30 patients enrolled between March 2013 and December 2014 were analyzed in this study.

Following mAb administration the bacterial count is unchanged (Fig 6P). This potential mechanism of "disease tolerance" should be addressed in detail in the discussion section.

Answer: We appreciate this reviewer bringing up this essential issue. In accordance with this review's request, we have made revision in the Discussion section (lines 313–330) as follows:

Our investigation revealed that excessive secretion of WARS1 represents a potential therapeutic target in severe hypercytokinemic sepsis. This condition, which leads to tissue parenchymal destruction and organ failure, is caused by a breakdown of disease tolerance that is not directly associated with immune resistance (i.e., pathogen load) (Soares *et al.*, 2017). This may explain why antimicrobial approaches are ineffective in treating diseases such as sepsis. Consequently, disease tolerance has been recognized as a reasonable pharmaceutical objective to control host stress and tissue damage, irrespective of the bacterial burden (Soares *et al.*, 2017). Injecting WARS1 protein into mice with mild sepsis resulted in substantial tissue damage, organ failure, and death, accompanied by the substantial production of proinflammatory cytokines and chemokines, which highlights the direct involvement of WARS1 in the pathophysiology of sepsis and its progression to severity. Anti-WARS1 MAb therapy, a novel pharmacological intervention, did not directly alter the bacterial load (Fig. 6P); however, it is thought to have preserved disease tolerance via its ability to inhibit the expression of multiple cytokines and chemokines at the infection site, thereby minimizing parenchymal cell damage (Chen *et al.*, 2018). The antibody treatment exerted anti-inflammatory effects via simultaneous inhibition of IL-6, CXCL8/IL-8, CCL3/MIP-1, TNF- α , and IFN- γ production, and had direct and considerable suppressive effects on excessive CCL and CXCL gene expression associated with sepsis immunopathogenesis.

Microphotographs of organ destruction are too small and do not allow for conclusive interpretation (Fig. 6)

Answer: According to this review's request, we have added larger microphotographs with better resolution in Fig 6.

Did you perform the animal experiments in a blinded fashion? Description is needed or also a justification for waiving this. Same is true for histological examination.

Answer: All the animal experiments and histological examinations were performed in a blinded manner. In accordance with the reviewer's comments, we have described the blinded examinations in the Materials and Methods section (lines 406–407 and 518–520).

Investigators were blinded to animal experimental procedures, including treatment, sample collection, and measurements.

Histological examination was performed in a blind manner by requesting a histopathologist from the Center of Animal Care and Use of Lee Gil Ya Cancer and Diabetes Institute.

Minor concerns

There are some semi-cola throughout the Manuscript, which do not improve the understanding (Abstract line 7 (71) and others).

Answer: We have corrected them in the revised manuscript (Abstract lines 71–74):

High plasma WARS1 levels stratified the early death of critically ill patients with sepsis, along with elevated levels of cytokines, chemokines, and lactate, as well as increased numbers of absolute neutrophils and monocytes, and higher Sequential Organ Failure Assessment (SOFA) scores.

N-terminal active domain, use the italic notation along current guidelines.

Answer: We have corrected it in the revised manuscript (lines 224–226).

As a therapeutic modality, we created an anti-human WARS1 monoclonal IgG1 antibody (anti-WARS1 MAb) with subnanomolar affinity for *N*-terminal active domain of human, marmoset, and mouse WARS1 (Fig EV4) (Ahn *et al.*, 2017).

Referee #3 (Comments on Novelty/Model System for Author):

The used methods are appropriate.

It is well known that the cytokine storm is a relevant feature of sepsis and septic shock associated with high mortality. The role of WARS1 in this respect is not fully understood. Several upstream mediators have been described that might be a useful target in treating sepsis.

So far, the results of applying anti-WARS1 indicate a beneficial effect during sepsis, however, how to identify the patients at risk and possibly benefiting from anti-WARS1 treatment? It is still unknown what possible effects the treatment with anti-WARS1 might have in respect to secondary viral infections or viral reactivation. This still needs to be answered in further studies.

The model system is adequate.

Referee #3 (Remarks for Author):

The authors investigated the role of WARS1 in sepsis using a translational approach and designed an anti-WARS1-neutralizing antibody showing beneficial effects during sepsis. The manuscript is nicely written.

Main questions:

How important is WARS1 in sepsis compared to septic shock? Although the low WARS1 group (Fig. EV2) shows no significant correlation with cytokines and chemokines, a still significant fraction dies from sepsis (Fig. 1C). Can the authors explain this experimentally and discuss in more detail when anti-WARS1 might be beneficial during sepsis? How robust would the discrimination into WARS1 low vs. high group be that identifies patients possibly benefiting from anti-WARS1 treatment?

Answer: We appreciate this reviewer's insightful queries. As WARS1 levels are higher in patients with septic shock than in those with sepsis, and injection of WARS1 into septic mice induces a cytokine and chemokine storm, increased lactate levels, and death, high levels of WARS1 appear to contribute to septic shock and death. However, our study could not provide any data concerning the deaths of sepsis patients with low WARS1 levels. As this reviewer

implied, we do not think that a specific WARS1 level at one time point can identify patients who would be beneficial from anti-WARS1 therapy. As we already mentioned in the Discussion section, the analysis of large patient samples and exploration of biomarkers and clinical parameters that can be used in conjunction with WARS1 would be required to achieve a more precise stratification for anti-WARS1 therapy. Although we acknowledge the importance of this question, a clear answer seems to require more extensive animal experiments and large size patient analysis.

Following this reviewer's suggestion, we have discussed this issue in the Discussion section (lines 303–309).

Nonetheless, considering the pathological complexity of sepsis, we do not think that measuring a specific WARS1 level at a given point would work as a sole determinant to stratify the patients who would respond to the anti-WARS1 MAb treatment. Systemic research should be followed involving the prospective analysis of multiple cohorts. For instance, how plasma WARS1 levels change over time in relation to the immunopathological state, disease severity, clinical parameters, biomarkers, and recovery or death would probably provide insights into the precise theranostic use of anti-WARS1 MAb.

We moved lines 334–336 to lines 289–291, and lines 337–343 to lines 309–312 in Discussion section.

Does anti-WARS1 treatment increases the risk for secondary viral infections during sepsis?

Answer: We appreciate this reviewer's question. Given that WARS1 is an innate immune activator against infection (Ahn *et al.*, 2017; Nguyen *et al.*, 2020), its depletion by long-term, continuous anti-WARS1 MAb treatment may raise the risk of opportunistic infection and/or subsequent viral infection. We think that extensive animal experiments are required to provide answers to this issue.

Further comments and questions:

WARS1 is secreted by monocytes. It has been shown that IL-3 is an upstream regulator of inflammation during sepsis, and that plasma IL-3 levels are positively correlated with circulating monocytes. Please provide data on the IL-3 levels in your sepsis cohorts. Is there a

correlation of IL-3 and WARS1? Can the authors discuss a possible link between IL-3, monocytes and WARS1, if a correlation can be detected

Answer: Although we currently do not have sufficient samples to measure IL-3 levels in patients with sepsis, our RNA-Seq data did not show a significant increase in IL-3 expression induced by WARS1, indicating the redundancy of upstream triggers, and differentiated signaling pathways. Further studies are required to examine the possible connections among IL-3, monocytes, and WARS1.

Can the authors subgroup the various monocyte population using CD14 and CD16 and perform correlation with WARS1?

Answer: It was previously reported that WARS1 levels are increased in CD16⁺ monocytes (Ancuta *et al*, 2009).

Do WARS1 levels correlate with monocyte apoptosis?

Answer: Unfortunately, we do not have any patient data showing the correlation between WARS1 levels and monocyte apoptosis.

I do not understand yet why the authors define WARS1 as upstream regulator. Please provide mechanistic data proving that WARS1 indeed induces the highly complex inflammatory signaling pathways leading to monocytosis, neutrophilia, and elevated cytokine and chemokine levels. So far the authors only provide correlative data. Of note, WARS1 is produced by monocytes, and higher WARS1 levels are correlated with higher monocytes and neutrophils, which are main producers of many of the mentioned cyto- and chemokines. But who induces high monocytes and neutrophils? IL-3?

Answer: Although we suggested that WARS1 could work as an upstream regulator, we are not sure at this moment whether it would be at the utmost position in the pathways. Further reciprocal studies on the relationship between WARS1 and IL-3 would help to elucidate the exact position of WARS1 in the signaling pathways.

Please provide apoptosis data on human monocytic THP-1 cells after rWARS1 treatment.

Answer: When THP-1 cells were treated with WARS1 (50 nM) for 6 and 24 h, cell viability (120%) increased significantly ($p < 0.001$). WARS1-induced monocyte apoptosis was not

observed under these experimental conditions.

What happens to the cytokine, chemokine and leukocyte levels when rWARS1 is injected i.v. into naive mice?

Answer: When rWARS1 was injected i.p. or i.v. into naïve mice, cytokine and chemokine levels were transiently and slightly elevated, and rapidly returned to normal levels. Unfortunately, we did not collect the leukocyte data.

What happens if anti-WARS1 is administered prior to LPS or CS injection?

Answer: Although we have not conducted the experiments raised by the reviewer, we think that anti-WARS1 MAb pretreatment could reduce innate inflammatory responses after LPS or CS injection.

References

- Ahn YH, Park S, Choi JJ, Park BK, Rhee KH, Kang E, Ahn S, Lee CH, Lee JS, Inn KS *et al* (2017) Secreted tryptophanyl-tRNA synthetase as a primary defence system against infection. *Nature Microbiology* 2
- Ancuta P, Liu KY, Misra V, Wacleche VS, Gosselin A, Zhou X, Gabuzda D (2009) Transcriptional profiling reveals developmental relationship and distinct biological functions of CD16+ and CD16- monocyte subsets. *BMC Genomics* 10: 403
- Chen L, Deng H, Cui H, Fang J, Zuo Z, Deng J, Li Y, Wang X, Zhao L (2018) Inflammatory responses and inflammation-associated diseases in organs. *Oncotarget* 9: 7204-7218
- Lu Z, Li Y, Jin J, Zhang X, Lopes-Virella MF, Huang Y (2012) Toll-like receptor 4 activation in microvascular endothelial cells triggers a robust inflammatory response and cross talk with mononuclear cells via interleukin-6. *Arterioscler Thromb Vasc Biol* 32: 1696-1706
- Mukaka MM (2012) Statistics corner: A guide to appropriate use of correlation coefficient in medical research. *Malawi Med J* 24: 69-71
- Nguyen TTT, Yoon HK, Kim YT, Choi YH, Lee WK, Jin M (2020) Tryptophanyl-tRNA Synthetase 1 Signals Activate TREM-1 via TLR2 and TLR4. *Biomolecules* 10
- Soares MP, Teixeira L, Moita LF (2017) Disease tolerance and immunity in host protection against infection. *Nat Rev Immunol* 17: 83-96

26th Oct 2023

Dear Prof. Jin,

Thank you for the submission of your revised manuscript to EMBO Molecular Medicine. We have now heard back from the two referees who we asked to re-evaluate your manuscript. As you will see from the reports below, while referee #2 supports publication of the manuscript, referee #3 acknowledges the improvements of the revised manuscript but remains critical particularly regarding, but not limited to, the lack of multivariate analysis for WARS1. Following points should be addressed in an additional and final round of major revision:

- Include the levels for TNF, CCL3, IFN γ and IL-8 in the existing patient characteristics tables (sepsis vs. septic shock and survivors vs. non-survivors).
- Perform a multivariate analysis for WARS1 including cyto- and chemokines, ANC, AMC, CRP, PCT as well as lactate for sepsis outcome.

As the above points could be addressed the existing data set, no additional experiments are required. Of course, any additional analysis and/or experiment that strengthen the main conclusion of the study are welcomed.

Further consideration of a revision that addresses reviewer's concerns in full will entail an additional round of review. Acceptance or rejection of the manuscript will depend on the completeness of your responses included in the next, final version of the manuscript. For this reason, and to save you from any frustrations in the end, I would strongly advise against returning an incomplete revision.

We would welcome the submission of a revised version within three months for further consideration. Please let us know if you require longer to complete the revision.

I look forward to seeing a revised form of your manuscript as soon as possible. Use this link to login to the manuscript system and submit your revision: Link Unavailable

I look forward to receiving your revised manuscript.

Yours sincerely,

Zeljko Durdevic

We require:

- 1) A .docx formatted version of the manuscript text (including legends for main figures, EV figures and tables). Please make sure that the changes are highlighted to be clearly visible.
- 2) Individual production quality figure files as .eps, .tif, .jpg (one file per figure). For guidance, download the 'Figure Guide PDF': (<https://www.embopress.org/page/journal/17574684/authorguide#figureformat>).
- 3) A .docx formatted letter INCLUDING the reviewers' reports and your detailed point-by-point responses to their comments. As part of the EMBO Press transparent editorial process, the point-by-point response is part of the Review Process File (RPF), which will be published alongside your paper.

- 4) A complete author checklist, which you can download from our author guidelines (<https://www.embopress.org/page/journal/17574684/authorguide#submissionofrevisions>). Please insert information in the checklist that is also reflected in the manuscript. The completed author checklist will also be part of the RPF.
- 5) Please note that all corresponding authors are required to supply an ORCID ID for their name upon submission of a revised manuscript.
- 6) It is mandatory to include a 'Data Availability' section after the Materials and Methods. Before submitting your revision, primary datasets produced in this study need to be deposited in an appropriate public database, and the accession numbers and database listed under 'Data Availability'. Please remember to provide a reviewer password if the datasets are not yet public (see <https://www.embopress.org/page/journal/17574684/authorguide#dataavailability>).
- In case you have no data that requires deposition in a public database, please state so in this section. Note that the Data Availability Section is restricted to new primary data that are part of this study.
- 7) For data quantification: please specify the name of the statistical test used to generate error bars and P values, the number (n) of independent experiments (specify technical or biological replicates) underlying each data point and the test used to calculate p-values in each figure legend. The figure legends should contain a basic description of n, P and the test applied. Graphs must include a description of the bars and the error bars (s.d., s.e.m.). See also 'Figure Legend' guidelines: <https://www.embopress.org/page/journal/17574684/authorguide#figureformat>

12) For more information: There is space at the end of each article to list relevant web links for further consultation by our

readers. Could you identify some relevant ones and provide such information as well? Some examples are patient associations, relevant databases, OMIM/proteins/genes links, author's websites, etc...

13) Author contributions: You will be asked to provide CRediT (Contributor Role Taxonomy) terms in the submission system. These replace a narrative author contribution section in the manuscript.

14) A Conflict of Interest statement should be provided in the main text.

Please note: When submitting your revision you will be prompted to enter your funding and payment information. This will allow Wiley to send you a quote for the article processing charge (APC) in case of acceptance. This quote takes into account any reduction or fee waivers that you may be eligible for. Authors do not need to pay any fees before their manuscript is accepted and transferred to the publisher.

EMBO Press participates in many Publish and Read agreements that allow authors to publish Open Access with reduced/no publication charges. Check your eligibility: <https://authorservices.wiley.com/author-resources/Journal-Authors/open-access/affiliation-policies-payments/index.html>

***** Reviewer's comments *****

Referee #2 (Remarks for Author):

After review of the revised version of this interesting MS I'm further impressed by the interesting actions of this interesting marker and mediator. Addition of the additional control group greatly improved the generalizability of conclusions from this work. All in all, the authors addressed all of my questions and minor/major issues in a very appropriate manner and revised the manuscript to improve its quality.

Referee #3 (Remarks for Author):

I thank the authors for explaining in more detail the raised issues. However, my main concerns have not been addressed so far. I would like to add: in the patient characteristics no significant difference is shown in survivors vs. non-survivors for the measured inflammatory parameters (CRP, PCT, ANC, AMC). Only for WARS1 and lactate. The levels of TNF, CCL3, IFN γ and IL-8 should be included in the patient characteristics (sepsis vs. septic shock and survivors vs. non-survivors) and not only showing it for WARS1high patients. This gives the reader a better sense regarding to the implied inflammatory conditions in these patient groups. By performing a multivariate analysis for WARS1 including the measured cyto- and chemokines, ANC, AMC, CRP, PCT as well as lactate would help to better understand the role of WARS1 during sepsis in humans, or if WARS1 is just a confounder of other significant parameters like lactate (e.g. how does the survival curve look like for lactate(high) vs. lactate(low) patients (similar to the WARS1(high) vs. WARS1(low) group?). This could be done with the existing data set.

Point by-point response to “EMM-2023-18106-V2”

We sincerely thank the reviewers for their efforts to improve our manuscript. We have tried to address reviewer #3's queries and hope that our response meets the reviewer's requirements. The revised manuscript has been annotated with red highlights.

Referee #2 (Remarks for Author):

After review of the revised version of this interesting MS I'm further impressed by the interesting actions of this interesting marker and mediator. Addition of the additional control group greatly improved the generalizability of conclusions from this work. All in all, the authors addressed all of my questions and minor/major issues in a very appropriate manner and revised the manuscript to improve its quality.

Answer: We really appreciate this reviewer's comments.

Referee #3 (Remarks for Author):

I thank the authors for explaining in more detail the raised issues. However, my main concerns have not been addressed so far. I would like to add: in the patient characteristics no significant difference is shown in survivors vs. non-survivors for the measured inflammatory parameters (CRP, PCT, ANC, AMC). Only for WARS1 and lactate. The levels of TNF, CCL3, IFN γ and IL-8 should be included in the patient characteristics (sepsis vs. septic shock and survivors vs. non-survivors) and not only showing it for WARS1 $_{high}$ patients. This gives the reader a better sense regarding to the implied inflammatory conditions in these patient groups. By performing a multivariate analysis for WARS1 including the measured cyto- and chemokines, ANC, AMC, CRP, PCT as well as lactate would help to better understand the role of WARS1 during sepsis in humans, or if WARS1 is just a confounder of other significant parameters like lactate (e.g. how does the survival curve look like for lactate $_{high}$ vs. lactate $_{low}$ patients (similar to the WARS1 $_{high}$ vs. WARS1 $_{low}$ group?)). This could be done with the existing data set.

Answer: We appreciate this reviewer's requests to provide valuable information for WARS1

as a biomarker for sepsis outcomes.

First, according to this reviewer's request, we added the levels of cytokine and chemokine, CXCL8/IL-8, CCL3/MIP-1 α , TNF- α , and IFN- γ , in Table EV1 and EV2, and below was described in the manuscript (lines 133–135).

The baseline characteristics of patients with sepsis and septic shock as well as survivors and non-survivors in 28-day mortality are shown in Table EV1 and EV2, respectively.

Table EV1. Baseline characteristics of sepsis and septic shock in the sepsis cohort

Characteristics	Sepsis (n=100)	Septic shock (n=89)	p -value
Age (years)	62 (54-71)	57 (47-68)	0.060
Sex (male), n (%)	64 (64)	63 (70)	0.350
SOFA score	8 (6-12)	16 (12-19)	<0.001
Mortality, n (%)	22 (22)	58 (65.1)	<0.001
Comorbidities, n (%)			
Respirology	10 (10)	4 (4.5)	0.170
Cardiology	34 (34)	24 (27)	0.340
Gastrology	9 (9)	6 (6.7)	0.600
Hepatology	16 (16)	31 (34.8)	0.004
Endocrinology	23 (23)	23 (25.8)	0.730
Renal	9 (9)	8 (9.0)	>0.99
Malignancy	49 (49)	57 (64)	0.040
Others	20 (20)	7 (7.8)	0.020
Laboratory findings			
ANC	9631 (5394-17445)	7950 (1885-16187)	0.070
ANC count, n (%)			
<500	10 (10)	18 (20)	
>500	90 (90)	71 (80)	
AMC	584 (340-947)	437 (128-796)	0.01
AMC count, n (%)			
<500	42 (42)	49 (55)	
>500	58 (58)	40 (45)	
Lactate (mmol/L)	1.6 (1.1-2.1)	5 (3-9)	<0.001
WARS1 (ng/mL)	50.1 (23.3-102.4)	82.6 (41.7-175.6)	<0.001

PCT (ng/mL)	1.1 (0.3-2.5)	1.6 (0.6-5.3)	0.002
CRP (ng/mL)	112.1 (70.6-155.6)	110 (51.9-162.2)	0.330
IL-8/CXCL8 (pg/mL)	32.3 (15.4-93.2)	260.0 (96.8-1142)	<0.001
CCL3/MIP-1 α (pg/mL)	9.9 (5.8-16.1)	18.6 (8.9-32.7)	<0.001
TNF- α (pg/mL)	18.5 (11.1-30.8)	28.0 (18.0-52.4)	<0.001
IFN- γ (pg/mL)	3.5 (1.6-7.9)	4.6 (2.2-9.3)	0.130
Site of infection, n (%)			0.690
Pneumonia	35 (35)	40 (44.9)	
Intraabdominal infection	25 (25)	24 (26.9)	
Urinary tract infection	12 (12)	3 (3.3)	
Others*	28 (28)	22 (24.7)	
Documented pathogens, n (%)			0.800
Gram-negative	37 (37)	37 (41.6)	
Gram-positive	24 (24)	22 (24.7)	
Mixed	10 (10)	6 (6.7)	
Anaerobes	0 (0)	3 (3.3)	
Others	1 (1)	0 (0)	
Unknown	28 (28)	21 (23.6)	
Bacteremia, n (%)	39 (39)	43 (48.3)	0.240

Table EV2. Baseline characteristics of survivors and non-survivors in the sepsis cohort

Characteristics	Survivors (n=109)	Non-survivors (n=80)	p-value
Age (years)	61 (47-69)	60 (49-67)	0.430
Sex (male), n (%)	74 (67.9)	53 (66)	0.880
SOFA score	9 (6.5-12.5)	16 (12-19)	<0.001
Diagnosis, n (%)			
Sepsis	78 (71.6)	22 (27.5)	
Septic shock	31 (28.4)	58 (72.5)	
Comorbidities, n (%)			
Respirology	12 (11)	2 (2.5)	0.050
Cardiology	36 (33)	22 (27.5)	0.430
Gastrology	8 (7.3)	7 (8.7)	0.790
Hepatology	18 (16.5)	29 (36.2)	0.002
Endocrinology	29 (26.6)	17 (21.2)	0.490
Renal	13 (11.9)	4 (5)	0.130

Malignancy	49 (44.9)	57 (71.2)	<0.001
Others	21 (19.3)	6 (7.5)	0.030
Laboratory findings			
ANC	9300 (5313-17435)	7411 (2159-16298)	0.120
ANC count, n (%)			
<500	12 (11)	16 (20)	
>500	97 (88.9)	64 (80)	
AMC	552 (291-880)	437 (141-845)	0.180
AMC count, n (%)			
<500	47 (43.1)	44 (55)	
>500	62 (56.9)	36 (45)	
Lactate (mmol/L)	1.7 (1.2-2.8)	5.3 (2.4-9.2)	<0.001
WARS1 (ng/mL)	53.3 (25.7-103.3)	82.9 (36.3-259.0)	0.007
PCT (ng/mL)	1.61 (0.39-3.41)	1.26 (0.39-4.65)	0.710
CRP (ng/mL)	113.2 (70.8-154.9)	105.6 (50.3-163.3)	0.240
IL-8/CXCL8 (pg/mL)	42.0 (16.8-164.8)	269.1 (55.6-1251.0)	<0.001
CCL3/MIP-1 α (pg/mL)	12.2 (6.2-24.8)	12.4 (8.0-30.2)	0.350
TNF- α (pg/mL)	22.0 (12.9-35.0)	24.5 (15.4-46.3)	0.120
IFN- γ (pg/mL)	4.6 (1.9-8.2)	3.4 (1.8-9.3)	0.640
Site of infection, n (%)			0.630
Pneumonia	33 (30.3)	42 (52.5)	
Intraabdominal infection	32 (29.3)	17 (21.2)	
Urinary tract infection	12 (11)	3 (3.7)	
Others*	32 (29.3)	18 (22.5)	
Documented pathogens, n (%)			0.790
Gram-negative	48 (44)	26 (32.5)	
Gram-positive	20 (18.3)	26 (32.5)	
Mixed	11 (10.1)	5 (6.2)	
Anaerobes	1 (0.9)	2 (2.5)	
Others	1 (0.9)	0 (0)	
Unknown	28 (25.7)	21 (26.2)	
Bacteremia, n (%)	48 (44)	34 (42.5)	0.880

Second, we conducted a multivariate analysis with the various parameters recommended by this reviewer for sepsis outcome. We described this statistical analysis in Materials and Methods Section lines 570–571, and the results were described in lines 142–143 as below. Among the parameters, lactate, IL-8/CXCL8, and WARS1 levels were risk factors associated with 28-day mortality. Our data are consistent with the previous reports on lactate (Jat *et al*, 2011; Oh *et al*, 2019) or IL-8/CXCL8 (Bozza *et al*, 2007; Kraft *et al*, 2015), respectively. We presented the data in Table EV3.

Additionally, multivariate analysis indicated that levels of WARS1 as well as lactate and IL-8/CXCL-8 were risk factors associated with 28-day mortality (Table EV3).

References

- Bozza FA, Salluh JJ, Japiassu AM, Soares M, Assis EF, Gomes RN, Bozza MT, Castro-Faria-Neto HC, Bozza PT (2007) Cytokine profiles as markers of disease severity in sepsis: a multiplex analysis. *Crit Care* 11
- Jat KR, Jhamb U, Gupta VK (2011) Serum lactate levels as the predictor of outcome in pediatric septic shock. *Indian J Crit Care Med* 15: 102-107
- Kraft R, Herndon DN, Finnerty CC, Cox RA, Song JQ, Jeschke MG (2015) Predictive Value of IL-8 for Sepsis and Severe Infections after Burn Injury: A Clinical Study. *Shock* 43: 222-227
- Oh DH, Kim MH, Jeong WY, Kim YC, Kim EJ, Song JE, Jung IY, Jeong SJ, Ku NS, Choi JY *et al* (2019) Risk factors for mortality in patients with low lactate level and septic shock. *J Microbiol Immunol Infect* 52: 418-425

Table EV3. Multivariate analysis for predicting 28-day mortality in the sepsis cohort

	Odds ratio	95% CI	p -value
Lactate	1.333	1.167–1.523	<0.001
IL-8/CXCL8*	1.006	1.001–1.010	0.023
WARS1*	1.026	1.001–1.052	0.039
PCT	0.896	0.797–1.007	0.065
AMC	1.000	0.999–1.001	0.070
CCL3/MIP-1 α	0.990	0.977–1.003	0.149
TNF- α	1.002	0.999–1.005	0.210
ANC	1.000	0.999–1.000	0.274
CRP*	1.015	0.959–1.076	0.593
IFN- γ	1.003	0.985–1.022	0.714

6th Nov 2023

Dear Prof. Jin,

Thank you for the submission of your revised manuscript to EMBO Molecular Medicine. I am pleased to inform you that we will be able to accept your manuscript pending the following final amendments:

1) Figures: All figures should be submitted as individual high-resolution files. Currently Figure 6 is uploaded in two parts. Please merge into one page and upload it as one file or split the figure into two separate figures (in that case update callouts in the text).

2) In the main manuscript file, please do the following:

- Please address all comments suggested by our data editors listed below:

o Please indicate the statistical test used for data analysis in the legends of figures 3a, d-g

o Please note that information related to n is missing in the legends of figures 1e-h; EV3a-c; EV4h-j; EV5b.

o Please note that the box plots need to be defined in terms of minima, maxima, centre, bounds of box and whiskers, and percentile in the legends of figures 7c-i

- Add up to 5 keywords.

- Please add callout for Fig 2 G-J. There is a callout for Fig 3J but there is no such panel in the figure 3. Please check and correct.

- Remove Table EV1-4 legends from the manuscript.

- In M&M, please include statement provided in the "Checklist" that the informed consent was obtained from all human subjects and that the experiments conformed to the principles set out in the WMA Declaration of Helsinki and the Department of Health and Human Services Belmont Report.

- Author contributions: Please remove it from the manuscript and specify author contributions in our submission system. CRediT has replaced the traditional author contributions section because it offers a systematic machine-readable author contributions format that allows for more effective research assessment. You are encouraged to use the free text boxes beneath each contributing author's name to add specific details on the author's contribution. More information is available in our guide to authors:

<https://www.embopress.org/page/journal/17574684/authorguide#authorshipguidelines>

3) Synopsis:

4) Source data: Please upload source data for one figure as one zipped file per figure and submit completed source data checklist.

5) As part of the EMBO Publications transparent editorial process initiative (see our Editorial at <http://embomolmed.embopress.org/content/2/9/329>), EMBO Molecular Medicine will publish online a Review Process File (RPF) to accompany accepted manuscripts. This file will be published in conjunction with your paper and will include the anonymous referee reports, your point-by-point response and all pertinent correspondence relating to the manuscript. Let us know whether you agree with the publication of the RPF and as here, if you want to remove or not any figures from it prior to publication. Please note that the Authors checklist will be published at the end of the RPF.

6) Please provide a point-by-point letter INCLUDING my comments as well as the reviewer's reports and your detailed responses (as Word file).

I look forward to reading a new revised version of your manuscript as soon as possible.

Yours sincerely,

Zeljko Durdevic

*** Instructions to submit your revised manuscript ***

1) a .docx formatted version of the manuscript text (including Figure legends and tables)

2) Separate figure files*

3) supplemental information as Expanded View and/or Appendix. Please carefully check the authors guidelines for formatting Expanded view and Appendix figures and tables at <https://www.embopress.org/page/journal/17574684/authorguide#expandedview>

4) a letter INCLUDING the reviewer's reports and your detailed responses to their comments (as Word file).

5) The paper explained: EMBO Molecular Medicine articles are accompanied by a summary of the articles to emphasize the major findings in the paper and their medical implications for the non-specialist reader. Please provide a draft summary of your article highlighting

This may be edited to ensure that readers understand the significance and context of the research.

Please refer to any of our published articles for an example.

6) For more information: There is space at the end of each article to list relevant web links for further consultation by our readers. Could you identify some relevant ones and provide such information as well? Some examples are patient associations, relevant databases, OMIM/proteins/genes links, author's websites, etc...

7) Author contributions: the contribution of every author must be detailed in a separate section.

8) EMBO Molecular Medicine now requires a complete author checklist (<https://www.embopress.org/page/journal/17574684/authorguide>) to be submitted with all revised manuscripts. Please use the checklist as guideline for the sort of information we need WITHIN the manuscript. The checklist should only be filled with page numbers where the information can be found. This is particularly important for animal reporting, antibody dilutions (missing) and exact values and n that should be indicated instead of a range.

9) Every published paper now includes a 'Synopsis' to further enhance discoverability. Synopses are displayed on the journal webpage and are freely accessible to all readers. They include a short stand first (maximum of 300 characters, including space) as well as 2-5 one sentence bullet points that summarise the paper. Please write the bullet points to summarise the key NEW findings. They should be designed to be complementary to the abstract - i.e. not repeat the same text. We encourage inclusion of key acronyms and quantitative information (maximum of 30 words / bullet point). Please use the passive voice. Please attach these in a separate file or send them by email, we will incorporate them accordingly.

You are also welcome to suggest a striking image or visual abstract to illustrate your article. If you do please provide a jpeg file 550 px-wide x 400-px high.

10) A Conflict of Interest statement should be provided in the main text

11) Please note that we now mandate that all corresponding authors list an ORCID digital identifier. This takes <90 seconds to complete. We encourage all authors to supply an ORCID identifier, which will be linked to their name for unambiguous name identification.

Currently, our records indicate that the ORCID for your account is 0000-0001-7268-3400.

Link Not Available

Photos 400-800 DPI

*Additional important information regarding figures and illustrations can be found at <https://bit.ly/EMBOPressFigurePreparationGuideline>. See also figure legend preparation guidelines: <https://www.embopress.org/page/journal/17574684/authorguide#figureformat>

***** Reviewer's comments *****

Referee #3 (Remarks for Author):

The authors have addressed all my comments.

The authors addressed the minor editorial issues.

10th Nov 2023

Dear Prof. Jin,

We are pleased to inform you that your manuscript is accepted for publication and is now being sent to our publisher to be included in the next available issue of EMBO Molecular Medicine.
